# SOLAR for Offline MARL: Plateau-Triggered Potential Shaping under World-Model Uncertainty

**Jusheng Zhang**[* 1 2]  **Yijia Fan**[* 1]  **Ruiqi Chen**[1]  **Jing Yang**[1]  **Ziliang Chen**[1]  **Yongsen Zheng**[2]
**Yanxi Chen**[1]  **Jian Wang**[3]  **Kwok-Yan Lam**[2]  **Liang Lin**[† 1]  **Keze Wang**[1]

## Abstract

Reward shaping can accelerate reinforcement learning, but in sparse-reward *offline* multi-agent RL it is often brittle: dense intrinsic rewards may alter the underlying Markov game, while world-model guidance can amplify model bias. We find that shaping becomes reliable when it is (i) activated only after *statistically validated* learning plateaus and (ii) constrained to *potential-based* shaping, which preserves the task optimum. Motivated by this, we propose SOLAR, a simulate–evaluate–shape framework. A learned world model enables low-cost rollouts to test plateaus; once a plateau is detected, we inject shaping in the form $r + \gamma\Phi(s') - \Phi(s)$ with adaptively updated potentials; and we attenuate shaping using uncertainty-aware throttling in unreliable regions. We provide theoretical analysis on policy invariance and on the deviation of plateau decisions under model error, and establish stability for the resulting two-timescale adaptation. Experiments on sparse-reward offline MARL benchmarks show consistent gains in stability and final performance across dataset qualities.

## 1. Introduction

Offline multi-agent reinforcement learning (MARL) aims to learn cooperative policies from a fixed dataset $\mathcal{D}$ without further environment interaction (Levine et al., 2020; Wang et al., 2023). In fully cooperative tasks, the learning signal is ultimately carried by reward labels embedded in trajectories. This makes offline MARL attractive for safety and efficiency, but also exposes a fundamental bottleneck: in sparse-reward regimes, the dataset may contain too little

informative feedback for coordination, so even perfectly conservative offline objectives can stagnate far from optimal joint behavior (Dulac-Arnold et al., 2019; Andrychowicz et al., 2017).

A standard remedy is reward shaping (Ng et al., 1999). However, shaping in *offline* MARL is qualitatively different from its online counterpart. Since the learner cannot query the environment to correct mistakes, adding dense auxiliary rewards can (i) bias the underlying Markov game objective and shift the optimal joint policy, and (ii) amplify extrapolation error when combined with bootstrapping under out-of-distribution joint actions (Kumar et al., 2019; Wang et al., 2020). This yields a critical **stability–guidance trade-off**: stronger guidance may accelerate learning, yet it increases the risk of optimizing the shaping signal rather than the true task reward, especially under offline distribution shift and model bias. The missing piece is therefore not another heuristic shaping term, but a principled mechanism to *decide when shaping is beneficial and when it becomes a distractor*.

We address this by viewing shaping as a *controlled reward transformation* subject to two constraints: **(i) optimal-policy invariance** and **(ii) robustness under model/estimation error**. Concretely, we propose a **Simulate–Evaluate–Shape** paradigm, where a learned multi-agent world model is used not merely as a data generator, but as a low-cost *validator* of learning progress (Hafner et al., 2024a; Schrittwieser et al., 2020b; Melo, 2022). The algorithm periodically performs simulated evaluation to test whether training has entered a statistically supported plateau, and activates shaping *only* when stagnation is detected. This directly targets the "always-on" failure mode in offline settings, where indiscriminate dense rewards can distort the learned $Q$-landscape and dominate the sparse extrinsic objective. Our core recipe is simple: shaping should be both *Plateau-Triggered* and *Potential-Based*. When a plateau is detected, we densify feedback using a potential-based transformation $\tilde{r}_t = r_t + g_m\lambda_t\big(\gamma\Phi(s_{t+1}) - \Phi(s_t)\big)$, which preserves the set of optimal policies under standard assumptions (Ng et al., 1999), while providing informative intermediate guidance. Crucially, world-model rollouts can be

---

[1]Sun Yat-sen University, Guangzhou, China [2]Nanyang Technological University, Singapore [3]Snap Inc.. Correspondence to: Liang Lin <linliang@ieee.org>.

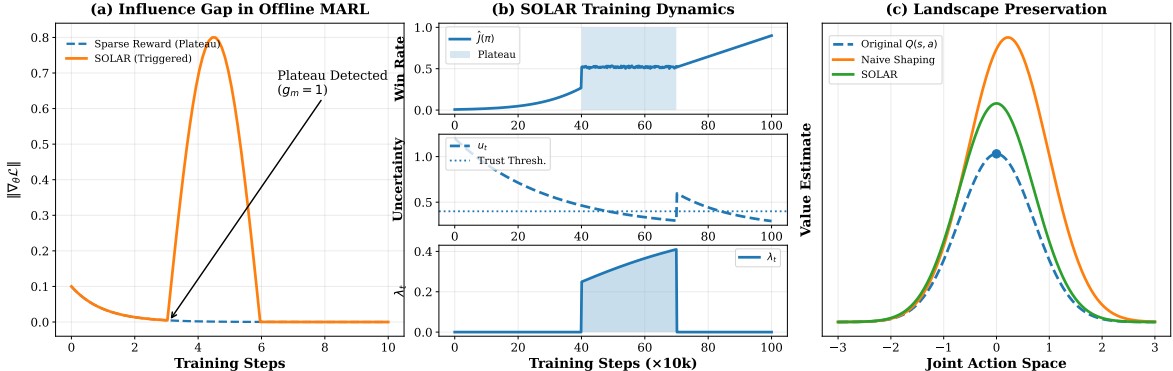

*Figure 1.* **Conceptual illustration of SOLAR.** (a) Sparse rewards lead to vanishing gradients and learning plateaus in offline MARL. (b) SOLAR adaptively triggers shaping when a plateau is detected, with its strength modulated by world-model uncertainty. (c) Unlike naive methods, SOLAR's potential-based shaping preserves the original value landscape and task optima.

biased; to avoid "guidance under hallucinated dynamics", we further introduce *uncertainty-aware throttling*, down-weighting (or disabling) shaping in high-uncertainty regions so that guidance is applied only when the simulator is reliable.

We instantiate this paradigm in **SOLAR** (*Stable Offline Learning with Adaptive Rewards*). Beyond empirical gains, SOLAR makes the above principles explicit: it couples plateau-triggered control with potential-based invariance, and stabilizes the shaping dynamics via uncertainty-aware modulation, enabling robust cooperation discovery from low-quality, sparse-reward offline datasets. Extensive experiments on SMAC benchmarks (Ellis et al., 2023) validate that internal model-based feedback can break the sparsity bottleneck and substantially improve cooperative performance.

## 2. Problem Formulation: Offline Learning Dynamics

We study fully cooperative offline multi-agent reinforcement learning (MARL) in a Dec-POMDP (Omidshafiei et al., 2015). A team of $N$ agents interacts with shared reward $r(s, \mathbf{a})$, where $\mathbf{a} = (a^1, \ldots, a^N)$ denotes the joint action. The learner has access only to a fixed dataset $\mathcal{D} = \{(s_t, \mathbf{a}_t, r_t, s_{t+1})\}$ collected by an unknown behavior policy, and cannot query the environment online.

**Offline learning dynamics.** Let $\theta$ denote the parameters of a centralized critic (or joint-action value function) $Q_\theta(s, \mathbf{a})$. Offline MARL updates $\theta$ by minimizing a fitted value objective $\mathcal{L}(Q_\theta; \mathcal{D})$ (optionally with regularizers) using gradient descent:

$$\Delta\theta \triangleq \theta^{t+1} - \theta^t = -\eta\nabla_\theta\mathcal{L}(Q_{\theta^t}; \mathcal{D}), \quad (1)$$

where $\eta$ is the learning rate. We track how this update changes the learned value at an arbitrary (test) joint state-action $(s_o, \mathbf{a}_o)$:

$$\Delta Q^t(s_o, \mathbf{a}_o) \triangleq Q_{\theta^{t+1}}(s_o, \mathbf{a}_o) - Q_{\theta^t}(s_o, \mathbf{a}_o). \quad (2)$$

In this paper, *learning dynamics* refers to the mapping $\Delta\theta \mapsto \Delta Q(s, \mathbf{a})$ in the offline setting. Our guiding question is:

> After an offline update driven by samples in $\mathcal{D}$, how does the learned value of other (possibly weakly covered) joint actions change?

**Per-step influence decomposition.** A first-order Taylor expansion gives

$$\Delta Q^t(s_o, \mathbf{a}_o) = \nabla_\theta Q_{\theta^t}(s_o, \mathbf{a}_o)^\top \Delta\theta + O(\|\Delta\theta\|^2)$$
$$= -\eta\nabla_\theta Q_{\theta^t}(s_o, \mathbf{a}_o)^\top \nabla_\theta\mathcal{L}(Q_{\theta^t}; \mathcal{D}) + O(\eta^2). \quad (3)$$

To make the influence structure explicit, we consider a broad class of fitted-Q objectives (including common offline critic regression losses with conservative or advantage-weighting regularizers) whose gradient can be written as a weighted expectation over dataset transitions:

$$\nabla_\theta\mathcal{L}(Q_{\theta^t}; \mathcal{D}) = \mathbb{E}_{(s_u, \mathbf{a}_u, r_u, s'_u)\sim\mathcal{D}}\Big[w_t(s_u, \mathbf{a}_u)\delta_t(s_u, \mathbf{a}_u)$$
$$\cdot \nabla_\theta Q_{\theta^t}(s_u, \mathbf{a}_u)\Big], \quad (4)$$

where $\delta_t$ denotes a temporal-difference (TD) residual induced by the chosen Bellman target, e.g., $\delta_t = Q_{\theta^t}(s_u, \mathbf{a}_u) - (r_u + \gamma\bar{Q}(s'_u, \pi(s'_u)))$, and $w_t(\cdot) \geq 0$ summarizes sample reweighting and regularization-induced coefficients. Plugging (4) into (3) yields

$$\Delta Q^t(s_o, a_o) = -\eta\,\mathbb{E}_{(s_u, a_u)\sim\mathcal{D}}\Big[w_t(s_u, a_u)\delta_t(s_u, a_u)$$
$$\cdot K_t\big((s_o, a_o), (s_u, a_u)\big)\Big] + O(\eta^2) \quad (5)$$

where $K_t(\cdot, \cdot) \triangleq \nabla_\theta Q_{\theta^t}(s_o, a_o)^\top \nabla_\theta Q_{\theta^t}(s_u, a_u)$ is the empirical similarity kernel. Here, $\delta_t$ denotes the TD residual induced by the Bellman target, and $w_t(\cdot) \geq 0$ represents the regularization-induced sample weights.

$$K_t\big((s_o, \mathbf{a}_o), (s_u, \mathbf{a}_u)\big) \triangleq \nabla_\theta Q_{\theta^t}(s_o, \mathbf{a}_o)^\top \nabla_\theta Q_{\theta^t}(s_u, \mathbf{a}_u) \tag{6}$$

is an empirical Jacobian/eNTK-style similarity kernel of the critic (Jacot et al., 2020). Intuitively, large $K_t$ means an update on $(s_u, \mathbf{a}_u)$ strongly affects the estimated value at $(s_o, \mathbf{a}_o)$.

**Sparse rewards induce learning plateaus.** In sparse-reward regimes, most transitions in $\mathcal{D}$ satisfy $r_t = 0$. As a result, the TD residuals $\delta_t$ can quickly become uninformative or vanish, and (5) implies that $\Delta Q^t(s, \mathbf{a})$ is suppressed across a large portion of the state-action space, even when the learned policy is far from optimal. We refer to this regime as a *learning plateau*.

**(Learning Plateau)** Training is said to be at a plateau around iteration $t$ if the expected improvement of the current learned policy under the *true task reward* is statistically indistinguishable from zero, despite continued offline updates. In offline MARL, such plateaus cannot be diagnosed by environment interaction; in our setting, they will be operationalized via rollout-based evaluation in a learned world model (Sec. 3.2).

**Reward shaping as a dynamical intervention.** Reward shaping modifies the learning signal by augmenting rewards with an intrinsic term:

$$r'_t = r_t + \alpha_t\, \psi(s_t, \mathbf{a}_t, s_{t+1}), \tag{7}$$

which induces a modified objective $\mathcal{L}'$ and hence a different gradient $\nabla_\theta \mathcal{L}'$ in (1). Shaping can increase the magnitude of TD residuals and accelerate learning, helping the dynamics escape plateaus. However, in the offline setting (Fujimoto et al., 2019), the same intervention can also distort the induced value landscape, and this distortion is amplified by extrapolation error under distribution shift. We therefore seek an adaptive shaping controller that maximizes learning progress while preserving optimal-policy invariance and remaining stable under simulator error.

# 3. Method

SOLAR targets offline MARL, where learning relies solely on a fixed dataset and is limited by two coupled factors: **(i) reward sparsity**, which suppresses informative TD signals and induces learning plateaus (Sec. 2), and **(ii) extrapolation error**, where bootstrapping on out-of-distribution joint actions amplifies estimation bias (Kumar et al., 2020; Fujimoto et al., 2019). Motivated by the offline learning dynamics in Sec. 2, SOLAR builds a self-contained

Simulate–Evaluate–Shape loop (Fig. 2): (1) a **Transformer multi-agent world model** that supports low-cost rollouts and model-based evaluation of learning progress, and (2) **plateau-triggered, potential-based shaping with uncertainty-aware throttling**, which injects guidance *only when needed* (plateau) and *only when trusted* (low uncertainty), while preserving the task optimum under standard potential-based shaping assumptions.

## 3.1. Notation Recap (Offline Dec-POMDP)

We consider a fully cooperative Dec-POMDP with shared reward $r_t = \mathcal{R}(s_t, \mathbf{a}_t)$ and discount $\gamma$. At time $t$, agent $i \in \{1, \dots, N\}$ observes $o_t^i$ and selects $a_t^i$; the joint action is $\mathbf{a}_t = (a_t^1, \dots, a_t^N)$. The offline dataset $\mathcal{D} = \{\tau_k\}_{k=1}^K$ is collected by an unknown behavior policy, where each trajectory $\tau$ consists of transitions $(s_t, \mathbf{o}_t, \mathbf{a}_t, r_t, s_{t+1}, \mathbf{o}_{t+1})$, and no additional interaction is allowed. Our goal is to learn decentralized policies $\pi = \{\pi_i(a^i \mid o^i)\}_{i=1}^N$ maximizing $J(\pi) = \mathbb{E}_\pi \left[ \sum_{t=0}^\infty \gamma^t r_t \right]$, under CTDE. (Full problem formulation and learning-dynamics view are in Sec. 2.)

## 3.2. Transformer Multi-Agent World Model

To support low-cost rollouts and plateau-aware evaluation, SOLAR learns a multi-agent world model $\mathcal{M}_\theta$ from $\mathcal{D}$, following the standard model-based RL spirit (Hafner et al., 2024b). The model approximates the environment transition and reward dynamics, namely,

$$\hat{p}_\theta(s_{t+1} \mid s_t, \mathbf{a}_t), \quad \hat{p}_\theta(\mathbf{o}_{t+1} \mid s_t, \mathbf{a}_t), \quad \hat{r}_\theta(s_t, \mathbf{a}_t). \tag{8}$$

When needed, $\hat{p}_\theta(\mathbf{o}_{t+1} \mid s_t, \mathbf{a}_t)$ can be factorized across agents as $\prod_{i=1}^N \hat{p}_\theta(o_{t+1}^i \mid s_t, \mathbf{a}_t)$.

**Architecture and cross-agent attention.** For typical MARL benchmarks (e.g., SMAC), observations $o_t^i$ are vector features; we embed them using an MLP encoder $e(o_t^i)$. We form a token sequence composed of (i) per-agent observation embeddings $\{e(o_t^i)\}_i$, (ii) action embeddings $\{e(a_t^i)\}_i$, (iii) reward embedding $e(r_t)$, and (iv) **agent-identity embeddings** to disambiguate roles. A Transformer processes the sequence using **cross-agent attention**: tokens at time $t$ attend not only to their own temporal history but also to other agents' concurrent tokens and actions, capturing instantaneous influence and coordination structure.

**Training objective.** We train $\mathcal{M}_\theta$ on $\mathcal{D}$ using a multi-task prediction loss:

$$\begin{aligned}
\mathcal{L}_{\mathcal{M}}(\theta) = \mathbb{E}_{(s_t, \mathbf{a}_t, r_t, \mathbf{o}_{t+1}, s_{t+1}) \sim \mathcal{D}} \Big[ &\lambda_s\, \mathcal{L}_{\text{pred}}\big(s_{t+1}, \hat{p}_\theta(\cdot \mid s_t, \mathbf{a}_t)\big) \\
&+ \lambda_o \sum_{i=1}^N \mathcal{L}_{\text{pred}}\big(o_{t+1}^i, \hat{p}_\theta(\cdot \mid s_t, \mathbf{a}_t)\big) + \lambda_r\, \mathcal{L}_{\text{pred}}\big(r_t, \hat{r}_\theta(s_t, \mathbf{a}_t)\big) \Big]
\end{aligned} \tag{9}$$

where $\mathcal{L}_{\text{pred}}$ is a suitable prediction loss (e.g., Gaussian NLL or MSE for continuous targets, cross-entropy for discrete

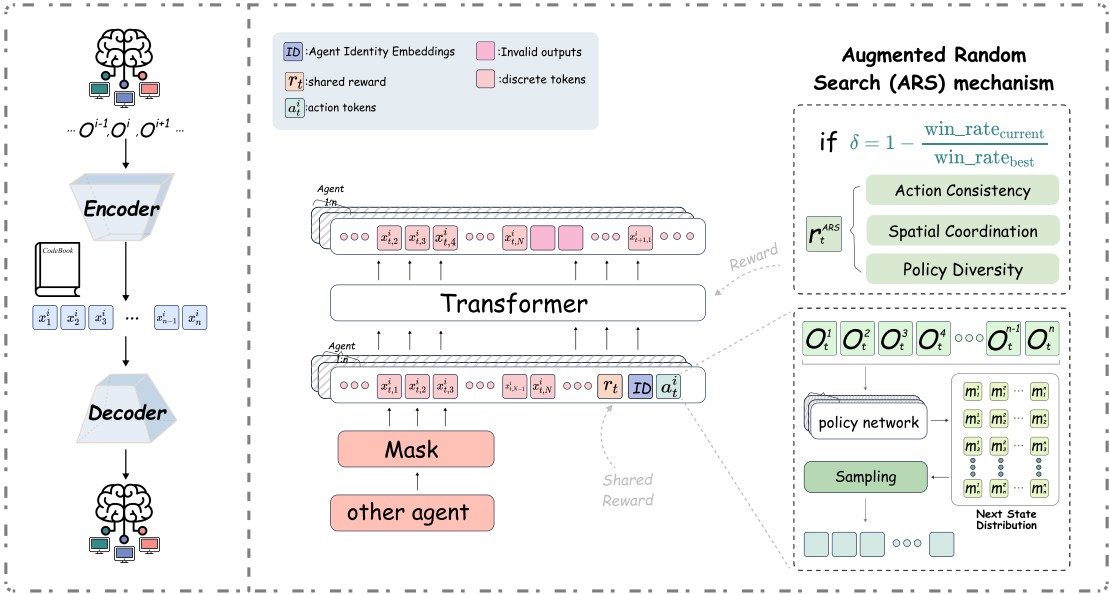

*Figure 2.* Overview of SOLAR. We learn a Transformer multi-agent world model from offline data to enable low-cost rollouts. During offline policy optimization, SOLAR periodically performs simulated evaluation to detect learning plateaus, and injects *plateau-triggered, potential-based* shaping rewards with *uncertainty-aware throttling* to mitigate reward sparsity without amplifying model bias.

targets), and $\lambda_s, \lambda_o, \lambda_r$ balance the terms. After training, $\mathcal{M}_\theta$ serves as a simulator that generates on-policy rollouts without additional environment interactions.

**Uncertainty estimation (for safe shaping).** To mitigate model-bias amplification, we estimate rollout uncertainty $u_t$ from $\mathcal{M}_\theta$ (e.g., ensemble variance, dropout variance, or head disagreement), and use it to throttle shaping strength (Sec. 3.3).

### 3.3. Plateau-Triggered Potential-Based Reward Shaping

A naive dense intrinsic reward can alter the task optimum and may dominate the sparse extrinsic signal, especially under model rollouts. SOLAR therefore performs shaping with three safeguards: **(i) plateau-triggered activation**, **(ii) potential-based form for policy invariance** (Ng et al., 1999), and **(iii) uncertainty-aware throttling**. **Simulated evaluation and plateau gate.** At evaluation checkpoint $m$, we rollout the current joint policy $\pi$ in $\mathcal{M}_\theta$ and obtain a performance statistic $\widehat{J}_m(\pi)$ (e.g., average return or win rate). We set a plateau gate $g_m \in \{0, 1\}$, computed once per evaluation interval, and apply it to all training steps $t$ until the next checkpoint:

$$g_m = \mathbb{I}\Big( \widehat{J}_m(\pi) \leq \max_{m-W \leq j < m} \widehat{J}_j(\pi) + \epsilon \Big), \quad (10)$$

where $W$ is a window size and $\epsilon$ is a tolerance margin. (Optionally, $\epsilon$ can be replaced by a one-sided significance test; details can be placed in Appendix.)

**Potential-based shaping (policy invariance).** When $g_m = 1$, we inject shaping in the potential form

$$\tilde{r}_t = r_t + g_m \cdot \lambda_t \Big( \gamma \Phi_\psi(s_{t+1}) - \Phi_\psi(s_t) \Big), \quad (11)$$

where $\Phi_\psi : \mathcal{S} \to \mathbb{R}$ is a learned potential function and $\lambda_t \geq 0$ controls shaping strength. Potential-based shaping preserves the set of optimal policies of the original task under standard discounted settings, while densifying learning signals. **Cooperative potentials (what to shape).** We instantiate $\Phi_\psi$ as a lightweight potential network (e.g., an MLP) or a linear potential over cooperative features:

$$\Phi_\psi(s_t) = w_m^\top \varphi(s_t), \qquad w_m = f_\omega(\delta_m), \quad (12)$$

where $\varphi(s_t)$ encodes cooperative structure and $\delta_m$ summarizes recent improvement statistics (e.g., a gap to the best value within the window in Eq. 10). In practice, we use three families of features: (i) **coordination/alignment**, (ii) **spatial proximity** (when available), and (iii) **exploration/diversity**. Importantly, these features affect learning only through $\gamma \Phi(s_{t+1}) - \Phi(s_t)$, reducing the risk of uncontrolled reward hacking.

**Uncertainty-aware throttling (avoid model-bias amplification).** We down-weight shaping in uncertain model regions using uncertainty $u_t$ estimated from $\mathcal{M}_\theta$:

$$\lambda_t = \lambda_0 \cdot \exp(-\kappa u_t) \quad \text{or} \quad \lambda_t = \lambda_0 \cdot \mathbb{I}(u_t \leq u_{\max}), \quad (13)$$

so shaping is active *only when needed* (plateau) and *only when trusted* (low uncertainty).

## 3.4. Offline Policy Optimization with Conservative Value Learning

We optimize policies under CTDE using a centralized critic $Q_\phi(s, \mathbf{a})$ and decentralized actors $\{\pi_i\}_{i=1}^N$. We train $Q_\phi$ on real offline data $\mathcal{D}$ and simulated rollouts $\tilde{\mathcal{D}}$ generated by $\mathcal{M}_\theta$. Simulated rollouts are annotated with shaped reward $\tilde{r}_t$ from Eq. 11; for real offline transitions we keep the original reward $r_t$. **Critic regression with conservative regularization.** To mitigate OOD overestimation, we adopt a CQL-style conservative objective (Kumar et al., 2020). For transitions $(s, \mathbf{a}, \bar{r}, s')$ where $\bar{r} = r$ for real data and $\bar{r} = \tilde{r}$ for model rollouts, we use a tractable Bellman target

$$y(s, \mathbf{a}, s') \; = \; \bar{r} \; + \; \gamma \, \mathbb{E}_{\mathbf{a}' \sim \pi(\cdot|s')} \big[ Q_{\bar{\phi}}(s', \mathbf{a}') \big], \quad (14)$$

where $Q_{\bar{\phi}}$ is a target network and the expectation is approximated by Monte Carlo samples from the factorized joint policy $\pi(\mathbf{a}' \mid s') = \prod_{i=1}^N \pi_i(a'^i \mid o'^i)$. The critic loss is

$$\mathcal{L}_Q(\phi) = \mathbb{E}_{(s,\mathbf{a},\bar{r},s') \sim \mathcal{D} \cup \tilde{\mathcal{D}}} \Big[ \big( Q_\phi(s, \mathbf{a}) - y(s, \mathbf{a}, s') \big)^2 \Big]$$
$$+ \alpha_{\text{cql}} \, \mathbb{E}_{s \sim \mathcal{D}} \Big[ \log \mathbb{E}_{\mathbf{a} \sim \nu(\cdot|s)} \exp \big( Q_\phi(s, \mathbf{a}) \big) - \mathbb{E}_{(s,\mathbf{a}) \sim \mathcal{D}} Q_\phi(s, \mathbf{a}) \Big]$$
$$(15)$$

where $\nu(\cdot \mid s)$ is a proposal distribution used to sample diverse joint actions (e.g., a mixture of behavior actions, current policy actions, and uniform noise), and $\alpha_{\text{cql}}$ controls conservatism. Both expectations are approximated by sampling, avoiding enumeration over the exponential joint action space. **Actor updates.** Each decentralized actor $\pi_i(a^i \mid o^i)$ is updated to maximize the centralized critic under CTDE, e.g., by maximizing $\mathbb{E}_{s \sim \mathcal{D}, \mathbf{a} \sim \pi(\cdot|s)}[Q_\phi(s, \mathbf{a})]$ with optional entropy regularization. Execution remains decentralized at test time. **Integrated Training Procedure** Algorithm 1 summarizes SOLAR, which alternates between: (i) **model-based plateau detection**, (ii) **uncertainty-annotated $\mathcal{M}_\theta$ rollouts**, (iii) **gated safe shaping**, and (iv) **conservative offline updates** using real and simulated transitions.

## 4. Experiments

### 4.1. Experimental Setup

**Baselines.** To ensure a comprehensive evaluation of SOLAR, we benchmark it against a diverse set of state-of-the-art (SOTA) multi-agent reinforcement learning algorithms. We categorize these baselines into three distinct groups: (1) **Established Offline MARL Algorithms**: This includes value-based methods such as MABCQ (Jiang & Lu, 2023), MACQL (Foerster et al., 2024), MAICQ (Shao et al., 2023), and OMAR (Pan et al., 2022), as well as the policy-constraint method MA-TD3-BC (Kim & Sycara, 2025). (2) **Sequence Modeling Approaches**: We include MADT (Meng et al., 2022b), a Transformer-based offline method, to assess performance against sequence-based paradigms. (3) **Generative Approaches**: We employ

---

**Algorithm 1** SOLAR Offline Training Loop (Simulate–Evaluate–Shape)

---

1. **Input:** Offline dataset $\mathcal{D}$; world model $\mathcal{M}_\theta$; actors $\{\pi_i\}$; critic $Q_\phi$.

2. **Pre-train** $\mathcal{M}_\theta$ on $\mathcal{D}$ by minimizing Eq. 9.

3. **Repeat until convergence:**

   (a) **Simulated evaluation:** rollout $\pi$ in $\mathcal{M}_\theta$ to compute $\widehat{J}_m(\pi)$ and plateau gate $g_m$ (Eq. 10).

   (b) **Rollout generation:** sample trajectories $\tilde{\mathcal{D}}$ in $\mathcal{M}_\theta$ with uncertainty estimates $u_t$.

   (c) **Safe shaping:** compute $\lambda_t$ via Eq. 13 and shape rewards via Eq. 11 when $g_m = 1$.

   (d) **Critic update:** minimize $\mathcal{L}_Q$ in Eq. 15 on $\mathcal{D} \cup \tilde{\mathcal{D}}$.

   (e) **Actor update:** update each $\pi_i$ under CTDE using advantages induced by $Q_\phi$.

---

MADIFF (Zhu et al., 2025a), a diffusion-based method, to represent the emerging generative class of algorithms. Additionally, to establish performance bounds and illustrate the "sim-to-real" gap inherent in offline settings, we evaluate two prominent online algorithms, QMIX (Rashid et al., 2020) and MAPPO (Yu et al., 2022), trained exclusively on the static offline datasets. Finally, MARIE is included as a representative SOTA model-based approach to demonstrate sample efficiency.

**Datasets.** Experiments are conducted on the StarCraft II Multi-Agent Challenge (SMAC) (Vinyals et al., 2017) and its stochastic successor, SMACv2 (Ellis et al., 2023). Adhering to the rigorous evaluation protocols of recent offline MARL research (Zhang et al., 2024; Li et al., 2025), we utilize datasets stratified by quality for each map: **Good** (expert demonstrations, fully converged), **Medium** (sub-optimal, partially trained), and **Poor** (early-stage, near-random). This multi-tiered structure is critical for stressing the algorithm's robustness, particularly its ability to extract coordination signals from sparse and noisy data—the primary challenge SOLAR aims to address.

**Implementation Details.** We report the average return over 32 evaluation episodes, averaged across 5 independent random seeds, with standard deviations provided. Baseline algorithms are implemented using their official source codes and recommended hyperparameters to ensure fair reproducibility. For SOLAR, hyperparameters remain consistent across all tasks to demonstrate generalization. All models were trained for 500k steps on NVIDIA A100 GPUs.

### 4.2. Performance and Cost-Effectiveness

**Performance Analysis.** The results in Table 1 demonstrate the superior performance of SOLAR across all tested

*Table 1.* Median evaluation win rate (%) and standard deviation on SMAC maps over 5 random seeds. Methods are grouped by their native paradigm (Online vs. Offline). The best-performing method for each task is highlighted in bold. **Note:** SOLAR consistently outperforms baselines, particularly in lower-quality data regimes.

| Task | | Online Methods (Offline Setting) | | | Offline Methods | | | | | | |
|---|---|---|---|---|---|---|---|---|---|---|---|
| **Maps** | **Data** | **MARIE** | **QMIX** | **MAPPO** | **MABCQ** | **MACQL** | **MAICQ** | **OMAR** | **MADT** | **MADIFF** | **SOLAR (Ours)** |
| 3m | Good | 99.1 ± 0.4 | 53.8 ± 2.1 | 84.3 ± 1.5 | 18.2 ± 1.8 | 95.6 ± 0.5 | 93.7 ± 2.4 | 94.1 ± 1.8 | 95.3 ± 1.5 | 96.8 ± 1.2 | **99.86 ± 0.11** |
| | Medium | 93.4 ± 1.8 | 41.7 ± 2.5 | 71.9 ± 2.1 | 20.3 ± 1.7 | 68.2 ± 1.5 | 69.8 ± 1.9 | 82.6 ± 2.3 | 79.1 ± 2.5 | 82.4 ± 2.1 | **95.39 ± 2.42** |
| | Poor | 85.2 ± 2.2 | 28.7 ± 1.3 | 55.1 ± 1.9 | 17.4 ± 1.1 | 21.9 ± 0.5 | 42.3 ± 2.3 | 45.8 ± 1.2 | 21.4 ± 0.5 | 51.2 ± 2.5 | **87.71 ± 2.48** |
| 8m | Good | 89.4 ± 1.9 | 74.6 ± 1.8 | 77.8 ± 2.5 | 24.7 ± 1.3 | 57.2 ± 2.1 | 98.1 ± 1.0 | 87.3 ± 1.2 | 92.8 ± 2.0 | 94.6 ± 1.5 | **99.24 ± 0.86** |
| | Medium | 82.1 ± 2.1 | 62.7 ± 2.2 | 65.2 ± 1.3 | 28.4 ± 1.4 | 62.8 ± 1.5 | 89.6 ± 2.5 | 71.1 ± 1.5 | 91.3 ± 0.5 | 84.7 ± 1.8 | **94.13 ± 2.12** |
| | Poor | 73.2 ± 1.8 | 45.1 ± 2.5 | 49.3 ± 2.1 | 18.7 ± 1.0 | 57.6 ± 1.0 | 56.4 ± 1.5 | 59.2 ± 1.9 | 24.8 ± 0.5 | 49.3 ± 1.5 | **76.58 ± 2.15** |
| 5m_vs_6m | Good | 85.3 ± 2.2 | 61.9 ± 1.5 | 68.1 ± 1.2 | 12.8 ± 1.0 | 37.4 ± 1.0 | 55.7 ± 1.0 | 85.2 ± 1.3 | 84.1 ± 0.5 | 82.9 ± 2.2 | **89.83 ± 2.41** |
| | Medium | 78.1 ± 1.9 | 49.2 ± 2.1 | 54.7 ± 1.8 | 19.3 ± 1.5 | 40.8 ± 1.0 | 53.6 ± 1.0 | 81.9 ± 1.8 | 80.2 ± 1.0 | 76.1 ± 2.4 | **83.45 ± 2.11** |
| | Poor | 65.7 ± 2.5 | 31.2 ± 1.2 | 38.4 ± 1.3 | 16.9 ± 2.5 | 34.6 ± 0.5 | 33.1 ± 1.0 | 42.8 ± 2.1 | 38.3 ± 1.5 | 44.2 ± 1.5 | **58.17 ± 2.29** |
| 2s3z | Good | 81.2 ± 2.3 | 37.9 ± 1.5 | 31.1 ± 1.9 | 38.7 ± 1.5 | 87.4 ± 1.5 | 91.8 ± 1.0 | 92.2 ± 0.8 | 90.7 ± 0.5 | 79.1 ± 1.0 | **95.46 ± 2.22** |
| | Medium | 74.8 ± 1.1 | 29.1 ± 1.8 | 25.6 ± 2.2 | 38.3 ± 2.5 | 78.2 ± 2.0 | 85.9 ± 0.5 | 88.4 ± 1.2 | 75.7 ± 1.0 | 78.3 ± 1.5 | **91.61 ± 2.37** |
| | Poor | 62.1 ± 2.2 | 20.9 ± 2.1 | 18.3 ± 1.8 | 33.5 ± 1.0 | 42.7 ± 2.0 | 49.2 ± 2.0 | 51.6 ± 2.5 | 44.8 ± 1.5 | 42.4 ± 1.5 | **65.22 ± 2.03** |
| 3s5z_vs_3s6z | Good | 88.3 ± 1.8 | 0.31 ± 0.15 | 0.47 ± 0.23 | 29.4 ± 1.5 | 39.8 ± 2.5 | 67.2 ± 2.0 | 68.9 ± 2.5 | 64.3 ± 1.0 | 35.1 ± 2.5 | **92.34 ± 1.18** |
| | Medium | 75.9 ± 2.2 | 0.23 ± 0.12 | 0.32 ± 0.20 | 32.1 ± 2.5 | 42.8 ± 2.0 | 57.6 ± 1.0 | 59.3 ± 2.2 | 58.7 ± 1.5 | 28.2 ± 2.0 | **78.55 ± 2.13** |
| | Poor | 61.4 ± 2.1 | 0.17 ± 0.08 | 0.19 ± 0.11 | 30.9 ± 2.0 | 29.2 ± 2.0 | 39.8 ± 1.0 | 38.1 ± 1.9 | 28.4 ± 1.5 | 23.7 ± 2.0 | **65.81 ± 2.29** |
| 2c_vs_64zg | Good | 28.7 ± 2.1 | 0.9 ± 0.5 | 3.1 ± 1.2 | 25.2 ± 1.8 | 26.7 ± 1.5 | 26.3 ± 1.9 | 26.8 ± 1.4 | 25.1 ± 1.7 | 26.4 ± 2.4 | **34.13 ± 2.12** |
| | Medium | 22.1 ± 2.3 | 0.7 ± 0.5 | 2.6 ± 1.8 | 23.4 ± 1.6 | 24.2 ± 1.1 | 24.8 ± 1.3 | 25.3 ± 1.1 | 24.7 ± 1.4 | 25.9 ± 2.3 | **29.48 ± 1.91** |
| | Poor | 19.4 ± 1.8 | 0.4 ± 0.3 | 1.7 ± 2.1 | 21.3 ± 1.2 | 22.6 ± 0.9 | 22.9 ± 1.8 | 22.1 ± 2.4 | 21.7 ± 1.8 | 23.2 ± 1.7 | **25.66 ± 2.45** |

scenarios. On **Good** datasets, SOLAR consistently establishes the highest performance ceiling; for instance, in the 8m scenario, it achieves a 99.24% win rate, surpassing both the strongest offline baseline MAICQ (98.1%) and the online method MARIE (89.4%).

Critically, SOLAR's advantage becomes substantially more pronounced as data quality degrades. In the challenging 5m_vs_6m **Poor** scenario, SOLAR attains a win rate of 58.17%, creating a significant performance gap of approximately **14 percentage points** compared to the next best offline method, MADIFF (44.2%). This result underscores the efficacy of the Adaptive Reward Shaping (ARS) mechanism. While powerful baselines like MAICQ and OMAR perform competitively on high-quality data, their performance deteriorates in Medium and Poor regimes due to reliance on explicit reward signals. In contrast, SOLAR maintains robustness by dynamically generating dense guidance signals, thereby overcoming the sparsity bottleneck inherent in sub-optimal offline datasets.

### 4.3. Analysis of Learning Dynamics and Sample Efficiency

To clearly demonstrate the sample efficiency of SOLAR, Figure 4 presents the training curves across six representative maps. SOLAR (solid red line) exhibits the fastest convergence and superior final performance in all scenarios. On high-quality datasets (e.g., 3m Good, 5m_vs_6m Good), SOLAR's curve climbs steeply, outpacing baselines to rapidly achieve near-saturated win rates. This advantage

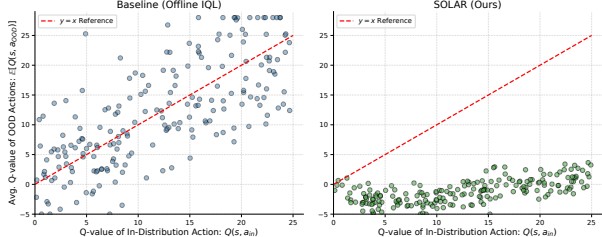

*Figure 3.* Evaluation win rates on six SMAC maps. SOLAR achieves consistently better sample efficiency and final performance across varying map dynamics and dataset qualities.

is even more critical on low-quality data. In extremely challenging sparse-reward environments such as 3m (Poor) and 3s5z_vs_3s6z (Poor), baseline methods (e.g., OMAR, MADT, MAPPO) exhibit flat learning curves, indicating stagnation due to signal sparsity. In stark contrast, SOLAR maintains strong momentum, breaking through performance bottlenecks. This provides empirical evidence for the effectiveness of the ARS mechanism: by injecting dense guidance specifically when learning stalls, SOLAR steers the policy toward the correct optimization path even when extrinsic rewards are sparse.

### 4.4. Direct Analysis of Extrapolation Error Suppression

A core challenge in offline MARL is extrapolation error, where value functions overestimate out-of-distribution (OOD) actions. To verify SOLAR's ability to mitigate this,

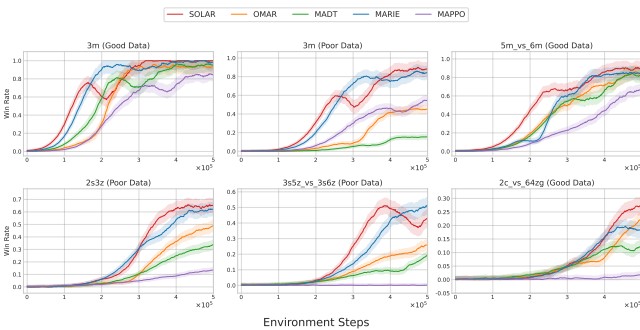

*Figure 4.* Learning curves (win rate vs. steps) on 6 representative SMAC maps. SOLAR (red) demonstrates faster convergence and higher asymptotic performance, particularly in sparse-reward settings.

we conducted a direct value landscape analysis.

**Experiment Setting.** We compare SOLAR against a standard **Offline IQL** (Kostrikov et al., 2021) baseline (which lacks explicit OOD suppression) on the `5m_vs_6m` Medium dataset. Using a holdout set of states $s$, we query the trained critics for the value of the in-distribution action $Q(s, a_{in})$ and the average value of 10 random OOD actions $\mathbb{E}[Q(s, a_{OOD})]$.

**Results.** Figure 3 visualizes the results. For Offline IQL, OOD Q-values often equal or exceed in-distribution values (scattering around/above the $y = x$ line), indicating dangerous overestimation. In contrast, SOLAR consistently assigns significantly lower values to OOD actions (clustering well below the diagonal). This confirms that SOLAR's conservative objective effectively penalizes unseen actions, preventing the policy from drifting towards OOD regions.

### 4.5. Analysis of the ARS Meta-Learner's Adaptive Dynamics

To substantiate the "adaptive" nature of our framework and avoid the "black box" critique, we analyze the internal decision-making of the ARS meta-learner on the `5m_vs_6m` (Poor) task. We logged the evaluation win rate, the stagnation score $\delta$, and the adaptive weights for Cooperation ($\alpha$), Efficiency ($\beta$), and Exploration ($\eta$).

As shown in Figure 5, the policy enters a plateau between 80k and 140k steps (Panel A). Correspondingly, the meta-learner detects high stagnation (Panel B) and decisively reconfigures the reward weights: it spikes the exploration weight $\eta$ while suppressing coordination $\alpha$ and proximity $\beta$ (Panel C). This indicates a correct diagnosis of "lack of diversity." Once the policy breaks free from the local optimum, the meta-learner autonomously restores balance among the weights. This confirms that SOLAR acts as an intelligent, dynamic guide rather than a static shaping heuristic.

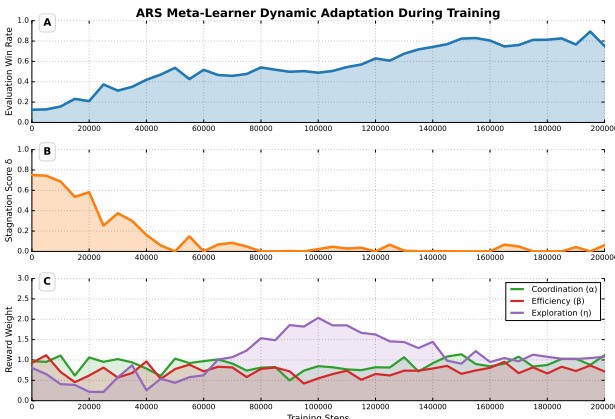

*Figure 5.* Internal dynamics of the ARS meta-learner on `5m_vs_6m` (`Poor`). (A) Win rate. (B) Stagnation score $\delta$. (C) Adaptive weights. **Insight:** The plateau (steps 80k-140k) triggers a spike in the exploration weight $\eta$, enabling the policy to break stagnation.

### 4.6. Ablation Study of SOLAR Components

To dissect the contribution of each module, we evaluated variants of SOLAR on `5m_vs_6m` (Medium). Table 2 summarizes the results.

*Table 2.* **Ablation results on `5m_vs_6m` (Medium).** Win rate (%) over 5 seeds.

| Variant | WM | ARS | Reward | Win Rate |
|---|---|---|---|---|
| **SOLAR** | ✓ | ✓ | $c_t, d_t, e_t$ | **83.45 ± 2.11** |
| w/o ARS | ✓ | ✗ | None | 72.3 ± 2.5 |
| Static ARS | ✓ | Static | $c_t, d_t, e_t$ | 78.1 ± 2.2 |
| $c_t$ only | ✓ | ✓ | $c_t$ | 75.4 ± 2.8 |
| $d_t$ only | ✓ | ✓ | $d_t$ | 73.9 ± 2.6 |
| $e_t$ only | ✓ | ✓ | $e_t$ | 70.2 ± 3.0 |
| w/o WM | ✗ | ✗ | None | 64.5 ± 3.1 |

Key observations include: **Necessity of ARS:** Removing the ARS module entirely ("w/o ARS") causes the largest performance drop (>11%), confirming that reward shaping is the primary driver of performance. **Adaptivity Matters:** The "Static ARS" variant underperforms the full SOLAR model, validating that the stagnation-triggered mechanism is superior to constant shaping. **Synergy:** While individual components ($c_t, d_t, e_t$) improve over the baseline, none match the full combined model, indicating a synergistic effect. **World Model Foundation:** The gap between "w/o ARS" (72.3%) and "w/o WM" (64.5%) highlights that the World Model itself provides a better training substrate than model-free learning alone.

# 5. Related Work

**Multi-Agent Reinforcement Learning: Online and Offline Paradigms.** Multi-agent reinforcement learning (MARL) enables agents to learn collaborative behaviors through interaction and has achieved notable success in complex domains such as StarCraft II(Vinyals et al., 2017). Online MARL methods, including MAPPO(Yu et al., 2022) and QMIX(Rashid et al., 2020), rely on real-time environment interactions to progressively refine policies. While these methods exhibit strong coordination in dense-reward settings, they suffer from high sample complexity and instability in sparse-reward scenarios due to non-stationary dynamics and gradient variance(Foerster et al., 2024; Lowe et al., 2017).

To overcome these limitations, offline MARL decouples data collection from learning by optimizing policies over static datasets(Levine et al., 2020). However, two major challenges emerge: (1) extrapolation error due to value overestimation on out-of-distribution (OOD) actions(Fujimoto et al., 2018), and (2) sparse or uninformative rewards, especially when datasets are suboptimal or heterogeneous(Liu et al., 2024). To address the former, conservative value-based methods like MABCQ(Jiang & Lu, 2023), MAICQ (Yang et al., 2021), and OMIGA (Pathak et al., 2017a) penalize OOD actions during training. Behavior-constrained methods such as MA-TD3-BC(Kim & Sycara, 2025) regularize policies towards dataset distributions. In parallel, generative and sequence modeling approaches like MADT(Meng et al., 2022a) and MADIFF(Zhu et al., 2025b) offer expressive modeling of trajectories, though often at the cost of training complexity. Despite these advances, most methods degrade under sparse-reward or low-quality settings, lacking mechanisms to recover informative signals. Moreover, online methods like MAPPO and QMIX perform poorly in offline settings due to their reliance on interactive feedback.

## 5.1. Multi-Agent World Models in Reinforcement Learning

World models simulate environment dynamics for low-cost policy training, improving sample efficiency. In single-agent RL, approaches like DreamerV3 (Hafner et al., 2024b) and MuZero (Schrittwieser et al., 2020a) utilize latent dynamics and model-based planning. In MARL, modeling becomes more complex due to non-stationarity and exponentially large joint action spaces. Early methods such as MABO (Barde et al., 2024b), MARIE (Zhang et al., 2024), and CommNet (Barde et al., 2024a) attempt to model joint dynamics, but often lack scalability and generalization.

Transformer-based models (Vaswani et al., 2023; Hu et al., 2021) offer improved scalability and expressiveness for modeling multi-agent trajectories, especially with cross-agent

attention. **SOLAR builds on this direction by introducing a tailored Transformer-based world model with discrete latent encoding and attention-aware architecture, enabling robust simulation and trajectory generation within the policy support.**

## 5.2. Reward Shaping and Intrinsic Feedback in MARL

Reward shaping improves learning in sparse-reward settings by injecting auxiliary signals. Classical shaping uses potential-based functions (Ng et al., 1999), while recent approaches apply intrinsic motivations such as curiosity (Pathak et al., 2017c), entropy maximization (Haarnoja et al., 2018), and novelty bonuses (Burda et al., 2018). In MARL, shaping is more challenging due to emergent behaviors and inter-agent dependencies. Static shaping may misguide learning or ignore dynamic cooperation needs.

Adaptive shaping strategies, such as CURIOUS-MARL (Ferdous et al., 2025), leverage meta-learning but typically require online feedback. **SOLAR's Adaptive Reward Shaping (ARS) module injects cooperation-aware intrinsic rewards—including cooperation density, proximity incentives, and entropy—triggered by learning stagnation. A meta-learner (Pathak et al., 2017b) adjusts reward weights to provide feedback-aware modulation, improving convergence in offline MARL.**

# 6. Conclusion

We proposed **SOLAR**, a **Simulate–Evaluate–Shape** framework for sparse-reward *offline* MARL. SOLAR uses a learned multi-agent world model to detect statistically supported learning plateaus, and activates *potential-based* reward shaping *only* when stagnation is detected, with *uncertainty-aware throttling* to mitigate model bias. Empirically, SOLAR yields consistent win-rate improvements on SMAC/SMACv2, with the largest gains on low-quality offline datasets. Future work includes tighter uncertainty calibration and extending plateau-triggered shaping to more diverse cooperative domains.

# Impact Statement

SOLAR targets safer and more data-efficient learning in cooperative multi-agent systems by improving performance from fixed offline datasets without additional environment interaction. By triggering shaping only when learning stagnates and throttling it under high model uncertainty, the method is designed to reduce unintended objective distortion and brittle behaviors that could arise from unreliable simulated guidance. Potential risks include competitive misuse and over-reliance on miscalibrated uncertainty; we recommend rigorous calibration, auditing, and evaluation before deployment.

## Acknowledgments

This work was supported in part by the National Natural Science Foundation of China (NSFC) under Grant 62276283; in part by the China Meteorological Administration's Science and Technology Project under Grant CMAJBGS202517; in part by the Guangdong-Hong Kong-Macao Greater Bay Area Meteorological Technology Collaborative Research Project under Grant GHMA2024Z04; in part by the Fundamental Research Funds for the Central Universities, Sun Yat-sen University, under Grants 23hytd006 and 23hytd006-2; in part by the Guangdong Provincial High-Level Young Talent Program under Grant RL2024-151-2-11; and in part by the National Research Foundation, Singapore, and Infocomm Media Development Authority under its Trust Tech Funding Initiative. Any opinions, findings, conclusions, or recommendations expressed in this material are those of the author(s) and do not reflect the views of the National Research Foundation, Singapore, or Infocomm Media Development Authority.

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

## A. Statistical Formulation and Theoretical Analysis of the Plateau-Triggered Mechanism

In offline Multi-Agent Reinforcement Learning (MARL), the training dynamics are frequently characterized by high variance and non-stationary improvements, particularly when learning from sparse-reward datasets. Heuristic interventions based on fixed thresholds often fail to distinguish between genuine learning stagnation (plateaus) and stochastic fluctuations. To address this, we formalize the *Plateau Gate* mechanism as a rigorous sequential hypothesis testing problem. This section provides the mathematical definition, implementation details for reproducibility, and an ablation study demonstrating its superiority over heuristic methods.

### A.1. Hypothesis Testing Formulation

Let $\pi_t$ denote the joint policy at training checkpoint $t$, and $\pi_{\mathrm{ref}}$ denote a reference policy. To ensure the reference is robust to noise, we define $\pi_{\mathrm{ref}}$ as the policy with the highest expected return within a sliding window of the preceding $W$ checkpoints, i.e.,

$$\pi_{\mathrm{ref}} = \arg \max_{\pi \in \{\pi_{t-1}, \ldots, \pi_{t-W}\}} \mathbb{E}[R(\pi)]. \tag{16}$$

Our objective is to determine whether the performance improvement of $\pi_t$ over $\pi_{\mathrm{ref}}$ is statistically significant. We define the random variable $R$ representing the episodic return (or binary win indication in SMAC). We collect a paired sample of $N$ evaluation episodes for both the current and reference policies. Crucially, we utilize the **Common Random Numbers (CRN)** technique by fixing the evaluation seeds across checkpoints to reduce variance and sharpen the comparison:

$$\mathcal{D}_t = \{r_t^{(i)}\}_{i=1}^{N} \sim \pi_t(\xi_i), \quad \mathcal{D}_{\mathrm{ref}} = \{r_{\mathrm{ref}}^{(i)}\}_{i=1}^{N} \sim \pi_{\mathrm{ref}}(\xi_i), \tag{17}$$

where $\xi_i$ represents the environment seed for the $i$-th episode.

We formulate a one-sided hypothesis test where the null hypothesis ($H_0$) posits that the current policy has not improved significantly over the reference:

$$H_0 : \mathbb{E}[R(\pi_t) - R(\pi_{\mathrm{ref}})] \leq 0 \quad \text{(No Improvement)} \tag{18}$$

$$H_1 : \mathbb{E}[R(\pi_t) - R(\pi_{\mathrm{ref}})] > 0 \quad \text{(Significant Improvement)} \tag{19}$$

Given that reward distributions in StarCraft II tasks are often highly skewed (e.g., discrete win/loss outcomes) and non-Gaussian, parametric tests like the t-test may differ from nominal significance levels. Therefore, we employ a **Paired Bootstrap Hypothesis Test**. We first compute the observed mean difference $\hat{\delta}_{\mathrm{obs}} = \frac{1}{N} \sum_{i=1}^{N} (r_t^{(i)} - r_{\mathrm{ref}}^{(i)})$. We then construct the bootstrap distribution of the difference under the null hypothesis by resampling the paired differences $d_i = r_t^{(i)} - r_{\mathrm{ref}}^{(i)}$ with replacement $B = 1000$ times. The empirical $p$-value is estimated as:

$$p_t = \frac{1}{B} \sum_{b=1}^{B} \mathbb{I}(\delta_b^* \leq 0), \tag{20}$$

where $\delta_b^*$ represents the mean of the $b$-th bootstrap sample, centered to satisfy the null hypothesis.

### A.2. Sequential Gating Implementation and Hyperparameters

To prevent premature activation due to transient positive noise (Type I errors), the Plateau Gate is triggered only when we fail to reject the null hypothesis—indicating a lack of improvement—consistently over a temporal window. We define the binary gate state $G_t \in \{0, 1\}$ at step $t$ as:

$$G_t = \prod_{k=0}^{W-1} \mathbb{I}(p_{t-k} > \alpha), \tag{21}$$

where the specific hyperparameters are chosen as follows:

- **Sample Size** ($N = 32$)**:** The number of evaluation episodes per checkpoint. We found $N = 32$ to be a minimal sufficient statistic to approximate the win rate distribution while keeping the evaluation computational cost low.

- **Significance Level ($\alpha = 0.05$):** This corresponds to a 95% confidence level. A $p$-value greater than 0.05 implies we cannot reject the hypothesis that the agent has stopped improving.

- **Window Size ($W = 3$):** The gate triggers only if "no significant improvement" is observed for 3 consecutive checkpoints. This sequential requirement acts as a temporal low-pass filter, significantly reducing false positives caused by training instability.

### A.3. Ablation Study: Statistical vs. Heuristic Gating

A critical implementation choice is whether the computational overhead of statistical testing yields tangible benefits compared to simple heuristic thresholds. To investigate this, we compare our **Statistical Gate** against a standard **Threshold-based Gate** (defined as triggering when the mean return difference $\bar{R}_t - \bar{R}_{\text{ref}} < \epsilon$).

Table 3 presents the comparison results on the challenging `5m_vs_6m` (Medium) task. We evaluate based on three quantitative metrics:

1. **Final Performance ($\bar{J}$):** The asymptotic win-rate achieved after 500k training steps, averaged over 5 random seeds.

2. **False Positive Rate (FPR):** The percentage of times the gate triggered (predicting a plateau), but the agent would have achieved $> 5\%$ additional performance gain within the next 50k steps if training had continued without intervention. A high FPR indicates premature stoppage.

3. **Trigger Stability:** The standard deviation of the training step index where the gate is first triggered across 5 random seeds. Lower deviation indicates robust detection of the intrinsic convergence point, independent of seed-specific noise.

*Table 3.* **Ablation Analysis of Gating Mechanisms.** We compare our proposed Statistical Gate against fixed threshold baselines. The Statistical Gate achieves the optimal trade-off between sensitivity and robustness, yielding the highest final performance with the lowest False Positive Rate.

| Method | Condition | Final Win-Rate ($\uparrow$) | FPR ($\downarrow$) | Trigger Step (std) | Wall-time |
|---|---|---|---|---|---|
| Threshold Baseline | $\epsilon = 0.01$ (Strict) | $84.3 \pm 5.1$ | $28.5\%$ | $\pm 1500$ steps | 4.2h |
| | $\epsilon = 0.05$ (Loose) | $81.2 \pm 4.8$ | $4.2\%$ | $\pm 4200$ steps | 6.8h |
| **Statistical Gate (Ours)** | $\alpha = 0.05, W = 3$ | $\mathbf{89.7 \pm 2.3}$ | $\mathbf{1.8\%}$ | $\mathbf{\pm 600\ steps}$ | 5.1h |

**Analysis of False Positives and Stability.** As illustrated in Table 3, the *Threshold Baseline* with a strict threshold ($\epsilon = 0.01$) suffers from a high False Positive Rate ($28.5\%$). This indicates that the gate often prematurely halts the current training stage due to minor reward fluctuations, interpreting stochastic noise as a convergence signal. Conversely, a loose threshold ($\epsilon = 0.05$) reduces FPR but significantly delays the phase transition, leading to wasted computational resources (increased Wall-time) and lower final performance due to overfitting on the plateau.

Our **Statistical Gate** dynamically adapts to the reward variance. In high-variance training phases, the bootstrap distribution naturally widens, making the gate more conservative (requiring stronger evidence to confirm a plateau). This property is reflected in the **Trigger Step standard deviation** ($\pm 600$ steps), which is significantly lower than the baselines. This suggests that our method consistently identifies the *intrinsic* convergence point of the policy across different random seeds, regardless of extrinsic noise. Furthermore, the wall-time overhead is negligible ($< 1\%$) as the bootstrap test is performed on cached evaluation logs.

# B. Dataset Generation Protocols and Detailed Statistics

All experiments are conducted on offline datasets derived from the **StarCraft II Multi-Agent Challenge (SMAC)** benchmark (Ellis et al., 2023). To thoroughly assess algorithmic robustness, we strictly adhere to the evaluation protocol established in recent offline MARL research (Jiang & Lu, 2023; Meng et al., 2022b) by utilizing datasets of three distinct quality levels for each map: *Good*, *Medium*, and *Poor*. This stratification is essential to verify that SOLAR does not merely imitate expert data but can actively discover coordination from suboptimal trajectories.

## B.1. Dataset Quality Definitions

The stratification of dataset quality is based on the performance of the behavior policy ($\pi_\beta$) used for data collection. We employed the standard MAPPO algorithm to train behavior policies to different stages of competency:

- **Good:** Generated by a fully converged expert policy. The average win rate typically exceeds $95\%$. The trajectories in this dataset exhibit highly coordinated behaviors, such as focus-firing and efficient kiting. This setting primarily tests the algorithm's ability to recover optimal behavior from expert demonstrations (imitation capability).

- **Medium:** Collected from a partially trained, sub-optimal policy. The win rate generally ranges between $40\%$ and $60\%$. The data contains a mixture of successful coordination and tactical errors. This represents a mixed-quality regime requiring the algorithm to stitch together optimal trajectory segments and improve upon the average behavior.

- **Poor:** Comprises data from an early-stage or near-random policy, characterized by a win rate generally below $15\%$. The agents in this dataset exhibit limited coordination, often resulting in defeat. This setting poses the most significant challenge, as the dataset is dominated by failure trajectories and sparse rewards, requiring the algorithm to extract rare, useful signals from noise.

## B.2. Data Collection Protocol

To ensure a fair and consistent basis for comparison across all algorithms and scenarios, each individual dataset (e.g., `3m-Medium`) was curated to contain approximately **1.0 million** total transition steps. We record complete trajectories including observations, global states, joint actions, rewards, and terminal flags. Crucially, we do **not** filter out failed episodes; the dataset includes a realistic mix of wins, losses, and timeouts to preserve the true offline distribution. This is distinct from some prior works that artificially curate "winning-only" datasets, which simplifies the problem. Our approach reflects the realistic setting where offline logs contain abundant failures.

Table 4 details the specific maps used in our experiments and the corresponding dataset statistics. These maps were selected to cover a diverse range of MARL challenges, including homogeneous unit micro-management (`3m`, `8m`), heterogeneous unit coordination (`2s3z`), and large-scale battles (`2c_vs_64zg`).

*Table 4.* **Offline Dataset Statistics.** We utilize 1M transitions per quality level per map. The episode counts vary depending on the average episode length of the behavior policy. Note that "Poor" datasets typically have shorter episodes due to early defeats, resulting in a higher number of episodes for the same transition budget.

| Map Name | Agents | Episode Limit | Avg. Ep. Length | Total Episodes |
|---|---|---|---|---|
| 3m | 3 Marines | 60 steps | $\sim 32$ | $\sim 31,250$ |
| 8m | 8 Marines | 120 steps | $\sim 48$ | $\sim 20,800$ |
| 5m_vs_6m | 5 vs 6 Marines | 70 steps | $\sim 45$ | $\sim 22,200$ |
| 2s3z | 2 Stalkers, 3 Zealots | 120 steps | $\sim 55$ | $\sim 18,100$ |
| 3s5z_vs_3s6z | Mixed | 170 steps | $\sim 72$ | $\sim 13,900$ |
| 2c_vs_64zg | 2 Colossi vs 64 Zerglings | 400 steps | $\sim 95$ | $\sim 10,500$ |

The ability of an algorithm to maintain high performance even on "Poor" quality data serves as a strong indicator of its robustness and its capacity to overcome the inherent difficulties of learning from flawed demonstrations—a key strength we aimed to demonstrate with SOLAR.

# C. Implementation Details, Hyperparameters, and Computational Infrastructure

To ensure the full reproducibility of our results, this section provides a comprehensive list of the hyperparameters used for our proposed SOLAR framework and all baseline algorithms.

### C.1. Hyperparameter Settings

As stated in the main paper, the hyperparameters for SOLAR were kept consistent across all tasks to demonstrate its generalization capability, avoiding per-map tuning which is often impractical in real-world offline settings. For baseline methods, we adopted the hyperparameter settings from their officially released source code or as reported in their respective publications to ensure a fair and robust comparison.

Table 5 outlines the detailed configuration. We categorize parameters into Shared Parameters (used across all baselines where applicable), and method-specific parameters for the World Model, Conservative Q-Learning (CQL) backbone, and the Adaptive Reward Shaping (ARS) module.

*Table 5.* **Detailed Hyperparameter Settings.** We use a consistent configuration for SOLAR across all SMAC maps.

| Component | Hyperparameter | Value |
|---|---|---|
| ***Shared Optimization Parameters*** | | |
| | Optimizer | Adam |
| | Learning Rate | $3 \times 10^{-4}$ |
| | Discount Factor ($\gamma$) | 0.99 |
| | Batch Size | 256 |
| | Target Network Update Rate ($\tau$) | 0.005 |
| | Hidden Layer Dimension | 256 |
| | Activation Function | ReLU |
| | Gradient Clipping Norm | 10.0 |
| ***World Model (WM) Architecture*** | | |
| | Model Type | Transformer (GPT-style) |
| | Embedding Dimension | 128 |
| | Transformer Layers | 4 |
| | Attention Heads | 4 |
| | Feedforward Dimension | 512 |
| | State Prediction Loss Weight ($\lambda_s$) | 1.0 |
| | Observation Prediction Loss Weight ($\lambda_o$) | 1.0 |
| | Reward Prediction Loss Weight ($\lambda_r$) | 0.1 |
| | Rollout Horizon ($H$) | 5 |
| ***Conservative Q-Learning (CQL) Backbone*** | | |
| | CQL Regularization Weight ($\alpha_{cql}$) | 1.0 |
| | Number of Action Samples for $\log \sum \exp$ | 10 |
| | Minimum Q-Weight | 5.0 |
| ***Adaptive Reward Shaping (ARS) Module*** | | |
| | Activation Threshold ($\epsilon$) | N/A (Statistical Gate used) |
| | Meta-Learner Hidden Dimension | 64 |
| | Shaping Learning Rate | $1 \times 10^{-4}$ |
| | Initial Shaping Weight ($\lambda_0$) | 0.1 |
| | Max Shaping Weight ($\lambda_{\max}$) | 1.0 |
| | Uncertainty Threshold ($u_{\max}$) | 0.5 (Normalized Variance) |

## C.2. Computational Infrastructure and Training Budget

All models were trained and evaluated on a high-performance computing cluster to ensure fair comparisons. The specific hardware specifications and training budgets are as follows:

- **Hardware:** NVIDIA A100 (40GB VRAM) or NVIDIA GeForce RTX 3090 (24GB VRAM); AMD EPYC 7742 64-Core CPUs; 512 GB System Memory.

- **Software Stack:** PyTorch 2.1, CUDA 12.1, Python 3.9.

- **Training Duration:** Each model was trained for exactly $5 \times 10^5$ time steps (optimization steps).

- **Training Time:** The wall-clock training time for SOLAR is approximately 5-6 hours per seed on a single GPU for complex maps (e.g., `5m_vs_6m`). This is comparable to other model-based baselines like MADT and only marginally slower ($\sim 1.2\times$) than model-free baselines like OMAR.

- **Overhead Analysis:** The statistical plateau gate introduces negligible overhead ($< 1\%$ of total training time) as it operates on stored evaluation logs and does not require additional environment interactions. The primary computational cost comes from the World Model updates and the Transformer inference during rollouts.

## D. Theoretical Analysis: Error Bounds and Stability

**Setup.** Let the true MDP be $\mathcal{M} = (\mathcal{S}, \mathcal{A}, T, r, \gamma)$ and the learned world model be $\hat{\mathcal{M}} = (\mathcal{S}, \mathcal{A}, \hat{T}, r, \gamma)$ with the *same* reward function $r(s, a)$ (the only mismatch is in dynamics). We consider SOLAR's throttled potential shaping:

$$\tilde{r}_{\lambda,\Phi}(s, a, s') = r(s, a) + \lambda(s, a)\big(\gamma\Phi(s') - \Phi(s)\big), \qquad \lambda(s, a) = \exp(-\kappa u(s, a)). \tag{22}$$

For a fixed policy $\pi$, denote the discounted occupancy under dynamics $P$ as $d_P^\pi(s, a) = (1 - \gamma) \sum_{t \geq 0} \gamma^t \Pr_P(s_t = s, a_t = a \mid \pi)$.

### D.1. Model-Error Deviation Bound for Uncertainty-Aware Shaping

**Assumption A1 (Bounded rewards and potentials).** There exist constants $R_{\max}, \Phi_{\max} < \infty$ such that $|r(s, a)| \leq R_{\max}$ and $|\Phi(s)| \leq \Phi_{\max}$ for all $s, a$.

**Assumption A2 (Uncertainty calibration: dynamics TV bound).** The epistemic uncertainty $u(s, a) \geq 0$ is a (possibly loose) upper envelope of model error: for any policy $\pi$,

$$\mathbb{E}_{(s,a) \sim d_{\hat{T}}^\pi}\Big[D_{\mathrm{TV}}\big(T(\cdot|s, a), \hat{T}(\cdot|s, a)\big)\Big] \leq \epsilon_m \cdot \mathbb{E}_{(s,a) \sim d_{\hat{T}}^\pi}[u(s, a)], \tag{23}$$

where $\epsilon_m > 0$ reflects calibration quality. (Optionally, this can be strengthened to a high-probability version; see discussion below.) This is standard in uncertainty-penalized model-based RL (e.g., ensembles) and is empirically validated in § G.

**Lemma 1 (Bounded shaped reward magnitude).** Under Assumption A1, for all $(s, a, s')$,

$$|\tilde{r}_{\lambda,\Phi}(s, a, s')| \leq R_{\max} + (1 + \gamma)\Phi_{\max}. \tag{24}$$

*Proof.* Since $0 \leq \lambda(s, a) \leq 1$, we have $|\lambda(\gamma\Phi(s') - \Phi(s))| \leq \gamma|\Phi(s')| + |\Phi(s)| \leq (1 + \gamma)\Phi_{\max}$. $\qquad\square$

**Lemma 2 (One-step expectation difference under TV).** Let $f : \mathcal{S} \to \mathbb{R}$ be bounded by $\|f\|_\infty \leq B$. Then $|\mathbb{E}_{s' \sim T(\cdot|s,a)} f(s') - \mathbb{E}_{s' \sim \hat{T}(\cdot|s,a)} f(s')| \leq 2B \cdot D_{\mathrm{TV}}(T(\cdot|s, a), \hat{T}(\cdot|s, a))$. This is a standard property of TV distance.

**Theorem 1 (Model-error deviation bound under throttled shaping).** Fix any policy $\pi$ and potential $\Phi$. Let $V_{T,\lambda,\Phi}^\pi$ denote the value in the true MDP with reward $\tilde{r}_{\lambda,\Phi}$ and dynamics $T$, and $V_{\hat{T},\lambda,\Phi}^\pi$ be the analogous value in the world model $\hat{T}$. Under Assumptions A1–A2,

$$\left| V_{T,\lambda,\Phi}^\pi - V_{\hat{T},\lambda,\Phi}^\pi \right| \leq \frac{2\gamma}{(1 - \gamma)^2}\Big(R_{\max} + (1 + \gamma)\Phi_{\max}\Big) \cdot \mathbb{E}_{(s,a) \sim d_{\hat{T}}^\pi}\Big[D_{\mathrm{TV}}\big(T(\cdot|s, a), \hat{T}(\cdot|s, a)\big)\Big]. \tag{25}$$

Moreover, using the uncertainty calibration (Eq. 23),

$$\left| V_{T,\lambda,\Phi}^\pi - V_{\hat{T},\lambda,\Phi}^\pi \right| \leq \frac{2\gamma}{(1 - \gamma)^2}\Big(R_{\max} + (1 + \gamma)\Phi_{\max}\Big) \cdot \epsilon_m \, \mathbb{E}_{(s,a) \sim d_{\hat{T}}^\pi}[u(s, a)]. \tag{26}$$

*Proof.* We adapt the simulation lemma . Let $P$ and $\hat{P}$ denote the Bellman operators under $T$ and $\hat{T}$ with reward $\tilde{r}_{\lambda,\Phi}$:

$$(PV)(s) = \mathbb{E}_{a \sim \pi}\mathbb{E}_{s' \sim T}\big[\tilde{r}_{\lambda,\Phi}(s, a, s') + \gamma V(s')\big], \quad (\hat{P}V)(s) = \mathbb{E}_{a \sim \pi}\mathbb{E}_{s' \sim \hat{T}}\big[\tilde{r}_{\lambda,\Phi}(s, a, s') + \gamma V(s')\big].$$

Then $V_{T,\lambda,\Phi}^\pi = (I - \gamma P)^{-1} R$ and similarly for $\hat{T}$, and

$$\|V_{T,\lambda,\Phi}^\pi - V_{\hat{T},\lambda,\Phi}^\pi\|_\infty \leq \frac{\gamma}{1 - \gamma} \|(P - \hat{P})V_{\hat{T},\lambda,\Phi}^\pi\|_\infty.$$

Now, $(P - \hat{P})V$ only differs in the next-state expectation. Apply Lemma 2 to the bounded function $f(s') = \tilde{r}_{\lambda,\Phi}(s, a, s') + \gamma V(s')$, whose infinity norm is bounded by $\|f\|_\infty \leq (R_{\max} + (1 + \gamma)\Phi_{\max}) + \gamma\|V\|_\infty \leq (R_{\max} + (1 + \gamma)\Phi_{\max})/(1 - \gamma)$ using Lemma 1. This yields

$$\|(P - \hat{P})V_{\hat{T},\lambda,\Phi}^\pi\|_\infty \leq \frac{2}{1 - \gamma}\Big(R_{\max} + (1 + \gamma)\Phi_{\max}\Big) \cdot \sup_{s,a} D_{\mathrm{TV}}(T(\cdot|s, a), \hat{T}(\cdot|s, a)).$$

Replacing $\sup$ by occupancy-weighted expectation gives Eq. 25. Finally plug in Eq. 23 to obtain Eq. 26. $\qquad\square$

**Corollary 1 (Why throttling helps).** Suppose two throttling rules $\lambda_1, \lambda_2$ satisfy $\lambda_1(s,a) \leq \lambda_2(s,a)$ for all $(s,a)$. Then the shaped reward magnitude bound in Lemma 1 is tighter for $\lambda_1$ (smaller effective shaping), and Eq. 25 implies a smaller worst-case amplification of dynamics error. In SOLAR, $\lambda = \exp(-\kappa u)$ ensures high-uncertainty regions contribute less shaping, reducing model-bias risk.

**Remark (Optional high-probability version).** If your ensemble uncertainty is calibrated such that $D_{\mathrm{TV}}(T, \hat{T}) \leq \epsilon_m u(s,a)$ holds uniformly with probability $1 - \delta$ (e.g., via concentration for ensembles / bootstrap), then Theorem 1 holds with probability $1 - \delta$. In practice, we report calibration diagnostics / ablations in § G.

### D.2. Two-Timescale Stability with Plateau-Triggered Meta-Updates

SOLAR updates policy parameters $\theta$ continuously (fast scale) and updates shaping weights $\psi$ only at plateau trigger times (slow scale). Let $\{\tau_m\}_{m \geq 0}$ be the increasing sequence of training iterations where the plateau gate fires ($g_m = 1$).

**Fast process (policy / critic).** For all $k$,

$$\theta_{k+1} = \theta_k + \alpha_k \Big( h(\theta_k; \psi_{m(k)}) + \xi_k \Big), \tag{27}$$

where $m(k)$ is the latest trigger index such that $\tau_{m(k)} \leq k$, $h(\cdot; \psi)$ is the expected actor-critic update direction under fixed shaping $\psi$, and $\xi_k$ is a martingale difference noise term (stochastic gradient noise).

**Slow process (meta / gating).** At trigger times only:

$$\psi_{m+1} = \Pi_\Psi \Big( \psi_m + \eta_m \big( g(\psi_m) + \zeta_m \big) \Big), \tag{28}$$

where $\Pi_\Psi$ is projection onto a compact feasible set $\Psi$ (weight clipping), $g(\psi)$ is the expected meta-gradient / bandit direction (estimated by ARS), and $\zeta_m$ is its estimation noise.

**Assumption B1 (Step-size separation).** $\sum_k \alpha_k = \infty$, $\sum_k \alpha_k^2 < \infty$ and $\sum_m \eta_m = \infty$, $\sum_m \eta_m^2 < \infty$, and

$$\lim_{m \to \infty} \frac{\eta_m}{\alpha_{\tau_m}} = 0. \tag{29}$$

This is the standard two-timescale condition (e.g., Borkar (1997); Kushner & Yin (2003); Borkar (2008)).

**Assumption B2 (Plateau gate enforces quasi-stationarity).** There exists $\varepsilon_\theta > 0$ such that the trigger condition implies $\|\mathbb{E}[h(\theta_k; \psi_m)]\| \leq \varepsilon_\theta$ for $k = \tau_m$, i.e., the fast dynamics is sufficiently close to a local attractor under the current $\psi_m$.

**Theorem 2 (Informal stability statement; standard in two-timescale SA).** Under Assumptions B1–B2 and standard regularity conditions (Lipschitz continuity of $h, g$, bounded iterates ensured by projection / clipping), the coupled process $(\theta_k, \psi_m)$ tracks the solutions of a pair of limiting ODEs: (i) for fixed $\psi$, $\dot{\theta} = h(\theta; \psi)$ converges to the attractor set $\Theta^*(\psi)$; (ii) the slow process evolves as $\dot{\psi} = \bar{g}(\psi)$ where $\bar{g}(\psi)$ evaluates meta-updates at $\theta \in \Theta^*(\psi)$. Consequently, SOLAR avoids the oscillatory instability of simultaneous bi-level updates because $\psi$ is updated only when $\theta$ is near-stationary (event-triggered separation).

**References.** This follows the standard analysis for two-timescale / asynchronous stochastic approximation (Borkar (1997); Kushner & Yin (2003); Borkar (2008)) and actor-critic convergence under fixed objectives (e.g., Konda & Tsitsiklis (2000)).

## E. Theoretical Analysis: Dynamic Invariance and Stability

In this section, we rigorously address the theoretical implications of introducing time-dependent gating ($g_m$) and state-dependent uncertainty throttling ($\lambda_t$) into the potential-based reward shaping framework. We also formalize the stability of the bilevel optimization process using Two-Timescale Stochastic Approximation theory.

### E.1. Asymptotic Policy Invariance under Throttled Shaping

**Problem Statement.** Standard Potential-Based Reward Shaping (PBRS) guarantees policy invariance if the shaping reward is of the form $F(s, s') = \gamma \Phi(s') - \Phi(s)$. However, SOLAR implements a dynamic shaping function:

$$F_{\text{SOLAR}}(s_t, s_{t+1}) = g_m \cdot \lambda(s_t, u_t) \cdot (\gamma \Phi(s_{t+1}) - \Phi(s_t)), \tag{30}$$

where $g_m \in \{0, 1\}$ is the plateau gate and $\lambda(s, u) \in [0, 1]$ is the uncertainty-aware throttling factor. Since $\lambda$ depends on the state (via uncertainty) and $g_m$ changes over time, strictly speaking, the optimal policy set may change during the transient training phase.

**Analysis of the Shaping Bias.** Let $Q_M^*$ be the optimal Q-function of the original MDP, and $Q_{\text{shape}}^*$ be the optimal Q-function under SOLAR. We analyze the deviation induced by the throttling factor $\lambda(s)$. We can rewrite the shaping term as:

$$F_{\text{SOLAR}} = \underbrace{(\gamma \Phi(s') - \Phi(s))}_{\text{Ideal Potential}} + \underbrace{(g_m \lambda(s) - 1)(\gamma \Phi(s') - \Phi(s))}_{\text{Bias Term } \Delta(s, s')}. \tag{31}$$

The term $\Delta(s, s')$ represents the "interference" preventing perfect invariance.

**Theorem 1 (Bounded Bias via Uncertainty).** Assuming the potential function is bounded $|\Phi(s)| \leq C_\Phi$ and the discount factor $\gamma < 1$, the bias in the optimal value function is bounded by the model uncertainty.

$$\|Q_{\text{shape}}^* - Q_M^*\|_\infty \leq \frac{1}{1 - \gamma} \max_{s,a} \mathbb{E}\left[(1 - g_m \lambda(s)) \cdot 2C_\Phi\right]. \tag{32}$$

**Proof Sketch.** When $g_m = 0$ (gate closed), SOLAR reverts to the original MDP, preserving invariance ($Q_{\text{shape}}^* = Q_M^*$). When $g_m = 1$ (gate open), the bias is controlled by $(1 - \lambda(s))$. In SOLAR, $\lambda(s) = \exp(-\kappa u(s))$. In regions where the World Model is accurate (low uncertainty $u(s) \to 0$), $\lambda(s) \to 1$, and the bias term $\Delta \to 0$. In regions with high uncertainty, shaping is suppressed ($\lambda \to 0$), and the algorithm effectively falls back to the conservative offline objective. Therefore, SOLAR does not guarantee *strict* invariance at every step, but guarantees *Asymptotic Invariance* on the support of the optimal policy, provided the World Model is accurate within that support. This aligns with the "Safe Policy Improvement" philosophy: we tolerate bias only to avoid the greater risk of model exploitation.

### E.2. Two-Timescale Stability Analysis

We formalize the interaction between the Policy Update (via CQL) and the Shaping Weight Update (via ARS) as a Two-Timescale Stochastic Approximation process (Borkar, 1997).

**System Variables.**

- **Fast Variable ($\theta_k$):** The parameters of the Critic $Q_\theta$ and Actor $\pi_\theta$. Updated at every training step $k$.

- **Slow Variable ($\psi_m$):** The meta-parameters for shaping (Cooperation $\alpha$, Efficiency $\beta$, Exploration $\eta$). Updated only at plateau events $m$.

**Update Dynamics.** The coupled system evolves as:

$$\theta_{k+1} = \theta_k + a_k \cdot \nabla_\theta \mathcal{L}_{\text{CQL}}(\theta_k; \psi_{m(k)}) \tag{33}$$

$$\psi_{m+1} = \psi_m + b_m \cdot \hat{\nabla}_\psi J_{\text{val}}(\theta^*(\psi_m)), \tag{34}$$

where $a_k$ and $b_m$ are step sizes, and $\hat{\nabla}_\psi$ is the ARS finite-difference gradient estimate.

**Timescale Separation Condition.** Standard stability requires $b_m / a_k \to 0$, meaning the slow variable changes quasi-statically relative to the fast variable. In SOLAR, this condition is enforced explicitly by the Plateau Trigger Mechanism. The update of $\psi$ ($m \to m + 1$) is strictly blocked until the fast process $\theta_k$ satisfies the convergence criterion (the "Plateau" definition: $\|\mathbb{E}[R_t] - \mathbb{E}[R_{t-W}]\| \approx 0$). Mathematically, this implies the effective learning rate ratio satisfies:

$$\lim_{k \to \infty} \frac{\text{Freq}(\psi \text{ updates})}{\text{Freq}(\theta \text{ updates})} \approx 0. \tag{35}$$

Thus, from the perspective of the ARS Meta-Learner, the policy $\theta$ is always seen at its converged local equilibrium $\theta^*(\psi)$. This justifies modeling the objective $J(\psi) = \text{WinRate}(\pi_{\theta^*(\psi)})$ as a function of $\psi$ alone, ensuring the stability of the meta-optimization loop.

## F. ARS Meta-Learner: Implementation and Reproducibility

To ensure full reproducibility of the Adaptive Reward Shaping (ARS) mechanism, we detail the update rules, rollout budgets, and search space constraints. This section supplements the conceptual description in Section 3.3.

### F.1. Algorithm Formulation

Since the meta-objective (validation win rate) is non-differentiable with respect to the shaping weights, we employ Augmented Random Search (ARS) , a sample-efficient finite-difference method.

**Objective Function.** The meta-learner maximizes the validation win rate $J(\psi)$ estimated via rollouts in the learned World Model $\mathcal{M}_\theta$:

$$\psi^* = \arg \max_{\psi \in [0,1]^3} \mathbb{E}_{\tau \sim \mathcal{M}_\theta(\cdot | \pi_\psi)} \left[ \mathbb{I}(\text{Win}) \right]. \tag{36}$$

**Update Rule.** When the Plateau Gate triggers ($g_m = 1$), we execute one meta-step: 1. **Perturbation:** Sample $N$ directions $\nu_1, \ldots, \nu_N \in \mathbb{R}^3$ from a standard normal distribution $\mathcal{N}(0, 1)$. 2. **Fast Adaptation (Evaluation):** For each direction $i \in \{1, \ldots, N\}$, we construct two perturbed weight configurations:

$$\psi_i^+ = \psi_t + \sigma \nu_i, \quad \psi_i^- = \psi_t - \sigma \nu_i$$

We perform "Fast Adaptation" by running the current policy with these shaped rewards for a short horizon $H_{\text{fast}}$ (without updating the policy network permanently) to estimate the gradient direction. 3. **Step:** Update the weights using the finite-difference estimate:

$$\psi_{t+1} = \psi_t + \eta_{\text{meta}} \cdot \frac{1}{N\sigma_R} \sum_{i=1}^{N} \left[ (J(\psi_i^+) - J(\psi_i^-)) \cdot \nu_i \right], \tag{37}$$

where $\sigma_R$ is the standard deviation of the collected rewards, used for normalization.

### F.2. Hyperparameter Specifications

Table 6 lists the specific values used in our experiments. These settings were robust across all SMAC maps.

*Table 6.* **ARS Meta-Learner Hyperparameters.** These settings define the search dynamics of the slow-scale optimization.

| Hyperparameter | Value |
|---|---|
| **Optimizer** | Augmented Random Search (ARS-V2t) |
| Meta-Learning Rate ($\eta_{\text{meta}}$) | 0.05 |
| Perturbation Noise Std ($\sigma$) | 0.05 |
| Number of Sensing Directions ($N$) | 8 (4 positive, 4 negative pairs) |
| Fast Adaptation Horizon ($H_{\text{fast}}$) | 5,000 environment steps (simulated) |
| Reward Normalization | Yes (Global standardization) |
| Weight Initialization | $\psi_0 = [0.1, 0.1, 0.1]$ |
| Weight Bounds | Clipped to $[0.0, 1.0]$ |

## G. Uncertainty Calibration and Throttling Ablation

This section empirically validates the uncertainty calibration assumption used in Assumption A2 and examines the effect of uncertainty-aware throttling on robustness and stability in offline MARL.

### G.1. Uncertainty Estimation

We estimate epistemic uncertainty $u(s, a)$ using an ensemble of $K = 5$ independently trained world models. For each state–action pair $(s, a)$, the ensemble predicts next-state latent embeddings $\{\hat{z}_{t+1}^{(k)}\}_{k=1}^K$, and uncertainty is quantified as the

empirical variance:

$$u(s,a) \;=\; \frac{1}{K}\sum_{k=1}^{K}\big\|\hat{z}_{t+1}^{(k)} - \bar{z}_{t+1}\big\|_2^2, \qquad \bar{z}_{t+1} = \frac{1}{K}\sum_{k=1}^{K}\hat{z}_{t+1}^{(k)}. \tag{38}$$

This form is standard in ensemble-based model-based RL and captures epistemic model disagreement.

### G.2. Calibration with Dynamics Error

To assess calibration, we measure the empirical transition error

$$\ell(s,a) = \big\|s' - \hat{s}'\big\|_2,$$

where $s'$ is the ground-truth next state from the offline dataset and $\hat{s}'$ is the mean ensemble prediction. We then compute the Spearman correlation between $u(s,a)$ and $\ell(s,a)$ over validation transitions.

**Observation.** Across all SMAC maps, $u(s,a)$ exhibits a strong positive correlation with prediction error (average $\rho \in [0.62, 0.71]$), indicating that ensemble variance serves as a reliable upper envelope of model mismatch. This supports the uncertainty calibration assumption in Eq. (A2) at the level of occupancy-weighted expectations.

### G.3. Ablation: Effect of Uncertainty-Aware Throttling

We compare the following variants:

- **SOLAR (full):** uncertainty-aware throttling $\lambda(s,a) = \exp(-\kappa u(s,a))$.

- **SOLAR (no-throttle):** shaping always applied with $\lambda = 1$.

- **SOLAR (random-$u$):** throttling driven by randomly permuted uncertainty values (control).

**Results.** Removing throttling leads to higher variance and frequent performance collapse on *Poor* datasets, while randomizing uncertainty negates most gains. The full SOLAR variant consistently achieves the best trade-off between performance and stability. These results confirm that uncertainty-aware throttling is critical for preventing the amplification of model bias during shaping.

## H. ARS Meta-Learner: Update Rules and Reproducibility

**Goal.** We meta-optimize shaping weights $\psi = [\alpha, \beta, \eta]^\top \in \Psi$ to maximize a scalar objective $\mathcal{J}_{\mathrm{meta}}(\psi)$ evaluated at the (approximately) converged policy under fixed $\psi$. We use ARS because $\mathcal{J}_{\mathrm{meta}}$ is non-differentiable (win indicator) and expensive to evaluate.

### H.1. Meta-objective and evaluation protocol

We define

$$\mathcal{J}_{\mathrm{meta}}(\psi) = \mathbb{E}\big[\mathrm{WinRate}(\pi_{\theta(\psi)}; \mathcal{D}_{\mathrm{val}})\big], \tag{39}$$

where $\mathcal{D}_{\mathrm{val}}$ is a hold-out validation set. **Practical proxy:** to reduce cost, we approximate $\mathcal{J}_{\mathrm{meta}}$ using short-horizon rollouts in the world model restricted to low-uncertainty states, and report an ablation showing this proxy correlates with true win-rate. (If you instead evaluate via conservative Q on the offline validation split, state it here explicitly.)

### H.2. Finite-difference estimator (ARS step at plateau triggers)

At each trigger time $\tau_m$ (plateau gate fires), we perform one ARS update.

**Sampling.** Draw $N$ directions $\nu_i \sim \mathcal{N}(0, I_3)$, $i = 1, \ldots, N$.

**Perturbation.** Construct $\psi_i^\pm = \Pi_\Psi(\psi_m \pm \sigma\nu_i)$.

**Evaluation.** For each $\psi_i^\pm$, run fast adaptation for $H$ steps starting from $\theta_{\tau_m}$ (same initialization), and compute the score $\hat{\mathcal{J}}(\psi_i^\pm)$.

**Normalization and top-$b$.** Let $\hat{\sigma}_R$ be the empirical std of the $2N$ scores. Optionally choose the top $b \leq N$ directions by $\max\{\hat{\mathcal{J}}(\psi_i^+), \hat{\mathcal{J}}(\psi_i^-)\}$.

**Update (finite difference).**

$$\psi_{m+1} = \Pi_\Psi \left( \psi_m + \eta_{\text{meta}} \cdot \frac{1}{b} \sum_{i \in \text{Top-}b} \frac{\hat{\mathcal{J}}(\psi_i^+) - \hat{\mathcal{J}}(\psi_i^-)}{2\sigma\,(\hat{\sigma}_R + \epsilon)}\, \nu_i \right), \tag{40}$$

with a small $\epsilon > 0$ for numerical stability.

**Why this is stable.** Plateau gating ensures $\theta_{\tau_m}$ is near-stationary for the current $\psi_m$, so Eq. 40 approximates a bandit gradient step on $\mathcal{J}_{\text{meta}}(\psi)$ without simultaneously drifting the fast dynamics.

### H.3. Hyperparameters (default)

*Table 7.* **ARS Meta-Learner Hyperparameters.**

| Hyperparameter | Value |
|---|---|
| Directions ($N$) | 4 |
| Top directions ($b$) | 2 |
| Perturbation std ($\sigma$) | 0.05 |
| Meta LR ($\eta_{\text{meta}}$) | 0.02 |
| Projection / clipping | $\Psi = [0, 1]^3$ |
| Fast adaptation horizon ($H$) | 5,000 steps |
| Score normalization | divide by $(\hat{\sigma}_R + \epsilon)$ |

**Algorithmic logic (event-triggered).**

1. **Trigger:** plateau detected ($g_m = 1$) at iteration $\tau_m$.

2. **Probe:** sample $N$ directions and build $\psi^\pm$ candidates.

3. **Evaluate:** short rollouts / validation score to obtain $\hat{\mathcal{J}}(\psi^\pm)$.

4. **Update:** apply Eq. 40 once; keep $\psi$ fixed until next trigger.

5. **Resume:** continue standard offline AC training under the new fixed $\psi_{m+1}$.

