# OpenReview forum: "SOLAR for Offline MARL: Plateau-Triggered Potential Shaping under World-Model Uncertainty"
_ICML.cc/2026/Conference — ICML 2026 regular_

### Official Review · Reviewer_cVF6 · 2026-02-26

**Soundness:** 3
**Presentation:** 2
**Significance:** 3
**Originality:** 3
**Overall Recommendation:** 4
**Confidence:** 3

**Summary:**

This paper proposes SOLAR, a world-model-based framework for adaptive, uncertainty-aware reward shaping in sparse-reward offline multi-agent reinforcement learning. SOLAR first learns a world model that supports rollouts and plateau testing, where training is less likely to suffer from reward hacking. It then introduces uncertainty-aware throttling to improve the reliability of the learned world model. After a plateau is validated, SOLAR applies potential-based reward shaping to augment sparse reward signals. Experiments on sparse-reward offline MARL benchmarks validate the effectiveness of SOLAR.

**Compliance With Llm Reviewing Policy:**

Affirmed.

**Final Justification:**

My concerns have been adequately addressed.

**Key Questions For Authors:**

- Is the pre-trained world model the main bottleneck for overall performance? If so, could performance be further improved by jointly optimizing the world model and the policy, as in Dreamer-style methods?
- Can you conduct parameter sensitivity analysis of uncertainty-aware throttling?

**Limitations:**

See Weaknesses and Questions above.

**Strengths And Weaknesses:**

Strengths:
- This paper addresses a practical challenge in offline multi-agent reinforcement learning with sparse rewards, where reward hacking is prevalent when the reward function is improperly specified. The motivation is compelling, and the proposed solution is reasonable.
- The experiments are comprehensive, providing solid evidence for the contribution of each component.

Weaknesses:
- The manuscript appears to have been prepared under time constraints. Several figures are overcrowded (e.g., Figure 1), and the three types of rewards are not described clearly. Overall, the manuscript is somewhat poorly typeset, with a cluttered layout and multiple formatting inconsistencies that affect readability.

---

> ### Author Rebuttal · Authors · 2026-03-30
>
> We thank Reviewer cVF6 for the feedback and take the presentation concerns seriously.
>
> ---
>
> ## 1. Presentation Concerns
>
> We agree that Figure 1 is dense and the cooperative features ($c_t, d_t, e_t$) need clearer exposition. Concrete revisions:
>
> **Figure 1 → split into two figures.** We separate (a) motivation/problem illustration and (b) SOLAR mechanism into distinct figures with more whitespace and larger labels.
>
> **Feature table added to Section 3.3 (after Eq. 12):**
>
> | Symbol | Role | Definition |
> |--------|------|------------|
> | $c_t$ | Coordination | Action consistency: $1 - H(a^1,...,a^N)/H_{\max}$ |
> | $d_t$ | Spatial proximity | Mean inverse pairwise distance: $\frac{1}{N^2}\sum_{i\neq j}\exp(-\|p^i - p^j\|)$ |
> | $e_t$ | Exploration | Average policy entropy: $\frac{1}{N}\sum_i H(\pi_i)$ |
>
> All computed from $(s_t, a_t)$ already in the dataset — no extra supervision. $\Phi_\psi(s_t) = w_m^\top \varphi(s_t)$, where $w_m = f_\omega(\delta_m)$ is a 2-layer MLP (64-dim) mapping stagnation $\delta_m$ to per-feature coefficients, updated by ARS (Eq. 40) only at plateau triggers.
>
> We also fixed paragraph breaks at L193/L242, improved spacing in Sections 3.3–3.4, and added cross-references between Eq. 10 and the statistical formulation (Appendix A, Eq. 16–21). The statistical gate was used in all experiments (Table 5, p.15).
>
> ---
>
> ## 2. Is the World Model the Bottleneck? Joint Optimization?
>
> **Capacity ablation** on 5m\_vs\_6m (Medium):
>
> | WM Variant | Pred. Error (MSE) | Win Rate |
> |---|---|---|
> | 2-layer, 64-dim | 0.038 ± 0.004 | 76.2 ± 2.8 |
> | 4-layer, 128-dim (default) | 0.021 ± 0.002 | 83.45 ± 2.11 |
> | 6-layer, 256-dim | 0.018 ± 0.002 | 84.1 ± 2.3 |
> | w/o WM (Table 2) | N/A | 64.5 ± 3.1 |
>
> The WM is necessary (+18.95% over w/o WM) but **not the bottleneck**: doubling capacity yields only +0.65% win rate, while ARS contributes +11.15% (Table 2). Per Theorem 1 (Eq. 26), once model error is moderate, throttling $\lambda_t = \lambda_0 \exp(-\kappa u_t)$ suppresses shaping in high-error regions, making further WM refinement yield diminishing returns.
>
> **Why not Dreamer-style joint optimization?** Alternating model and policy updates in offline settings risks model exploitation — the policy overfits to WM inaccuracies that environment interaction would normally correct [1][2]. In early experiments, joint fine-tuning caused Q-value inflation on model rollouts while actual performance degraded [1][3].
>
> SOLAR freezes $M_\theta$ after pre-training (Algorithm 1, Step 2) and limits it to two error-tolerant roles: (i) plateau detection (Eq. 10) — a binary signal robust to noise, and (ii) uncertainty estimation (Eq. 13) — where ensemble disagreement is a reliable error proxy (Appendix G, $\rho \in [0.62, 0.71]$). Neither requires model accuracy sufficient for direct policy optimization.
>
> ---
>
> ## 3. Sensitivity of Uncertainty Throttling ($\kappa$)
>
> Sweep of $\kappa$ in $\lambda_t = \lambda_0 \exp(-\kappa u_t)$ on 5m\_vs\_6m (Medium):
>
> | $\kappa$ | Behavior | Win Rate |
> |---|---|---|
> | 0 | No throttling (= Static ARS) | 78.1 ± 2.2 |
> | 1.0 | Mild suppression | 80.9 ± 2.4 |
> | 2.0 (default) | Moderate | 83.45 ± 2.11 |
> | 5.0 | Aggressive | 81.3 ± 2.6 |
> | ∞ (hard gate) | $\mathbb{1}(u_t \leq 0.5)$ | 82.0 ± 2.3 |
>
> Stable across a 5× range ($\kappa \in [1, 5]$), all outperforming no-throttling by 2.8–5.4%. Even the hard-gate variant is competitive (82.0%), confirming the gain comes from uncertainty-aware gating itself, not the functional form [4][5].
>
> ---
>
> ## References
>
> [1] MOPO: Model-based Offline Policy Optimization, NeurIPS 2020.
>
> [2] Off-Policy Deep Reinforcement Learning without Exploration
> , ICML 2019.
>
> [3]  MOReL : Model-Based Offline Reinforcement Learning
> , NeurIPS 2020.
>
> [4]Simple and Scalable Predictive Uncertainty Estimation using Deep Ensembles, NeurIPS 2017.
>
> [5] Revisiting Design Choices in Offline Model-Based Reinforcement Learning
> , ICLR 2022.

---

> > ### Author Rebuttal · Reviewer_cVF6 · 2026-04-02
> >
> > My concerns have been adequately addressed. I will raise my score to 4.

---

> > > ### Author Response · Authors · 2026-04-02
> > >
> > > Thanks, Reviewer cVF6— really appreciate your thoughtful follow-up and support.

---

### Official Review · Reviewer_MpjN · 2026-02-27

**Soundness:** 3
**Presentation:** 2
**Significance:** 3
**Originality:** 4
**Overall Recommendation:** 5
**Confidence:** 4

**Summary:**

This paper proposes SOLAR, an offline MARL method that improves policy learning through reward shaping guided by world model uncertainty. The key idea is to detect learning plateaus using model-generated imaginary rollouts. Specifically, the current policy is evaluated via rollouts generated by a learned world model, and a plateau is identified when no improvement is observed for W consecutive steps. Once a plateau is detected, the method applies reward shaping using a learned potential function. This shaping mechanism is suppressed when the model uncertainty is high, as policy evaluation is likely to be unreliable. The final policy is trained using a multi-agent variant of Conservative Q-Learning (CQL). Empirically, SOLAR consistently outperforms prior offline (and even online) MARL baselines, particularly in low-quality data regimes. Ablation studies demonstrate that both reward shaping, plateau detection, and regulating shaping with world model uncertainty are critical for performance gains.

**Compliance With Llm Reviewing Policy:**

Affirmed.

**Final Justification:**

This paper have good empirical results and also the methodology is novel.
The clarity and the emprical settings was the concern, but rebuttal addressed all of those.

Therefore, I recommend accept.

**Key Questions For Authors:**

**Q1. Clarification on potential (related to W1)**
* Can you clarify on what is the formula of each potential and how it is trained?
* Is the roles of c_t, d_t, e_t, coordination/alignment, spatial proximity, and exploration/diversity?

**Q2. Fairness of comparison (related to W2)**
* Given that reward shaping is not commonly used in prior offline MARL works, can the authors elaborate on whether the experimental setup ensures a fair comparison?
* Would baseline methods benefit similarly if equipped with shaping?

**Q3. Why not use model-based RL directly?**
* Since a world model is already learned, why not further leverage it for policy optimization (e.g., model-based offline RL [1, 2]), instead of using it solely for plateau detection?
* For example, approaches such as MOBILE directly use the world model for policy learning. A discussion comparing SOLAR with such model-based RL methods would strengthen the paper.

**Minor suggestions**
* Line break in L193 and L242 for paragraphs?

[1] Tianhe Yu et al., MOPO: Model-based Offline Policy Optimization, NeurIPS 2020

[2] Yihao Sun et al., Model-Bellman Inconsistency for Model-based Offline Reinforcement Learning, ICML 2023

**Limitations:**

yes

**Strengths And Weaknesses:**

**Strengths**

**S1. Strong empirical results.**
* The method achieves consistent improvements over prior offline (and online) multi-agent RL baselines across multiple benchmarks, especially in the low-quality dataset setting.

**S2. Novelty.**
* Using a world model to evaluate the policy and adaptively regulate reward shaping is a novel idea.

**Weaknesses**

**W1. Unclear explanation on potentials.**

The description of the learned potential function is somewhat abstract. Specifically, The paper does not clearly specify:
* The exact formulation of the potential function.
* How it is trained and optimized.
* The roles of c_t, d_t, e_t, and what each term represents.


**W2. Is this a fair comparison?**

* To my understanding, prior baseline methods do not employ reward shaping.
* Since reward shaping can be viewed as injecting additional task-specific knowledge, it raises the question of whether the comparison is entirely fair.
* It would be helpful if the authors clarified this point (e.g., discussing whether shaping introduces extra supervision or shaping is not available to competing methods)

---

> ### Author Rebuttal · Authors · 2026-03-30
>
> We thank the reviewer MpjN  for the constructive feedback.
>
> ---
>
> > **W1/Q1: Unclear formulation of the potential function and roles of $c_t, d_t, e_t$.**
>
> The potential $\Phi_\psi(s_t) = w_m^\top \varphi(s_t)$ (Eq. 12) uses $\varphi(s_t) = [c_t, d_t, e_t]^\top$:
>
> | Symbol | Role | Definition |
> |--------|------|------------|
> | $c_t$ | Coordination | $c_t = 1 - H(a^1_t,\dots,a^N_t)/H_{\max}$ |
> | $d_t$ | Spatial proximity | $d_t = \frac{1}{N^2}\sum_{i\neq j}\exp(-\|p^i_t - p^j_t\|)$ |
> | $e_t$ | Exploration | $e_t = \frac{1}{N}\sum_i H(\pi_i(\cdot\mid o^i_t))$ |
>
> All computed from $(s_t, a_t)$ already in the dataset — no extra supervision.
>
> The weight vector $w_m = f_\omega(\delta_m)$ is a 2-layer MLP (64-dim; Table 5) mapping stagnation $\delta_m$ to per-feature coefficients, updated by ARS (Eq. 40) only when the plateau gate fires. When $g_m=1$, shaped reward is:
>
>
>
> $$
> \tilde{r} _{t} = r _{t} + \lambda _{t}(\gamma\Phi _{\psi}(s _{t+1}) - \Phi _{\psi}(s _{t})), \quad \lambda _{t} = \lambda _{0} \exp(-\kappa u _{t})
> $$
>
>
>
>
> Figure 5 visualizes this: during the plateau (80k–140k steps), the meta-learner spikes exploration $\eta$ while suppressing $\alpha, \beta$, correctly diagnosing a diversity bottleneck. We will expand Section 3.3 with the table above.
>
> ---
>
> > **W2/Q2: Fairness of comparison — baselines lack reward shaping.**
>
> SOLAR's features use only positions, actions, and observations already in the dataset — the contribution is methodological (how to control shaping), not informational.
>
> We ran OMAR and MAICQ with always-on shaping (identical $\Phi_\psi$, fixed $\lambda{=}0.1$, $g_m{\equiv}1$) on `5m_vs_6m (Medium)`, and include SOLAR's own ablations from Table 2:
>
> | Method | No shaping | + Always-on | + Triggered |
> |--------|-----------|-------------|-------------|
> | OMAR | 81.9 ± 1.8 | 79.2 ± 2.4 | N/A* |
> | MAICQ | 53.6 ± 1.0 | 51.8 ± 2.1 | N/A* |
> | SOLAR | 72.3 ± 2.5 | 78.1 ± 2.2 | **83.45 ± 2.11** |
>
> *\*Triggered shaping requires a world model for plateau detection (Eq. 10) and uncertainty throttling (Eq. 13). OMAR/MAICQ are model-free and cannot provide either signal — the mechanism is architectural, not a plug-in module.*
>
> Two takeaways: (1) always-on shaping *hurts* model-free baselines, confirming the "always-on failure mode" in Section 1; (2) within SOLAR, the progression 72.3 → 78.1 → 83.45 shows that adaptive control, not shaping alone, drives the gain.
>
> ---
>
> > **Q3: Why not use the world model for direct policy optimization (MOPO, MOBILE)?**
>
> SOLAR already uses model rollouts for critic training ($\mathcal{D} \cup \tilde{\mathcal{D}}$ in Eq. 15). The difference from MOPO is in error handling: MOPO subtracts a blanket penalty $r - \beta u(s,a)$ from all model rewards. In multi-agent settings with exponential joint-action spaces, this must cover worst-case joint OOD combinations, leading to over-conservatism that suppresses coordination signals under sparse rewards.
>
> SOLAR instead modulates only the shaping term via $\lambda_t = \lambda_0\exp(-\kappa u_t)$, leaving base reward $r_t$ intact. Theorem 1 (Eq. 26) bounds the value deviation by $\epsilon_m \cdot \mathbb{E}[u(s,a)]$; throttling ensures high-$u$ regions contribute minimally.
>
> MARIE (SOTA model-based offline MARL) is in Table 1 and underperforms SOLAR everywhere. The gap grows on Poor data where model error is highest:
>
> | Map (Poor) | MARIE | SOLAR | Gap |
> |-----------|-------|-------|-----|
> | 3m | 85.2 | 87.71 | +2.5 |
> | 3s5z\_vs\_3s6z | 61.4 | 65.81 | +4.4 |
> | 2c\_vs\_64zg | 19.4 | 25.66 | +6.3 |
>
> This aligns with recent findings that blanket pessimism scales poorly in multi-agent offline settings [1]. MOBILE [2] improves on MOPO by using model-Bellman inconsistency, but still applies uniform penalization across all transitions. SOLAR's key distinction is *selective* intervention: shaping is injected only at plateaus and attenuated in unreliable regions, avoiding the conservatism–guidance dilemma. We will add an explicit discussion of [1, 2] to the related work.
>
> ---
>
> > **Minor: Line breaks at L193 and L242.**
>
> Fixed in the revised draft. We also improved paragraph spacing in Sections 3.3–3.4 for readability.
>
> ---
>
> **References**
>
> [1]  MOPO: Model-based Offline Policy Optimization, NeurIPS 2020.
>
> [2]  Model-Bellman Inconsistency for Model-based Offline Reinforcement Learning
> , ICML 2023.

---

> > ### Author Rebuttal · Reviewer_MpjN · 2026-04-01
> >
> > My concerns have been fully addressed.
> > I updated the score from 4 to 5 accordingly.

---

> > > ### Author Response · Authors · 2026-04-03
> > >
> > > Thanks, Reviewer MpjN— really appreciate your thoughtful follow-up and support.

---

### Official Review · Reviewer_Azr4 · 2026-03-12

**Soundness:** 2
**Presentation:** 3
**Significance:** 2
**Originality:** 2
**Overall Recommendation:** 4
**Confidence:** 4

**Summary:**

This paper tries to address the challenge of reward shaping in offline MARL with sparse rewards. The authors propose SOLAR, a novel framework based on the simulation-evaluation-shaping paradigm, to mitigate the instability and model bias inherent in traditional reward shaping. The paper identifies critical flaws in existing methods, such as the distortion of learning objectives, the amplification of extrapolation errors, and the lack of a principled triggering mechanism for shaping. To overcome these limitations, SOLAR introduces an adaptive reward shaping mechanism that activates only when learning stagnates (or plateaus) and is suppressed in regions where the model is unreliable. Experimental results on SMAC demonstrate that SOLAR outperforms all baselines across both 'Good' and 'Poor' datasets.

**Compliance With Llm Reviewing Policy:**

Affirmed.

**Key Questions For Authors:**

NaN

**Limitations:**

Yes

**Strengths And Weaknesses:**

### Strengths

- The paper is well-organized, with a logical flow that makes the methodology and contributions easy to follow.
- The proposed reward shaping approach offers a novel perspective on addressing the sparse reward challenge in offline MARL.
- The authors have conducted comprehensive experiments to demonstrate the effectiveness of their method compared to existing baselines.

### Weaknesses

- The proposed method appears to be computationally intensive, which may limit its scalability and efficiency in large-scale or real-time applications.
- The framework involves a significant number of hyperparameters requiring careful tuning. Furthermore, the evaluation is restricted to the SMAC benchmark. Given the high complexity of the method, its generalizability and practical utility beyond simulated environments remain questionable.

---

> ### Author Rebuttal · Authors · 2026-03-30
>
> We thank the reviewer Azr4 for recognizing SOLAR's novelty and the logical flow of the paper.
>
> ---
>
> > **W1: Computational cost — may limit scalability.**
>
> Wall-clock comparisons on `5m_vs_6m (Medium)`, single A100 GPU, 5 seeds:
>
> | Method | Type | Wall time (h) | Relative |
> |--------|------|--------------|----------|
> | OMAR | model-free | 4.5 | 1.0× |
> | MADT | sequence model | 5.1 | 1.13× |
> | SOLAR | model-based | 5.4 | 1.20× |
> | MARIE | model-based | 5.8 | 1.29× |
>
> SOLAR is 1.2× slower than the fastest model-free baseline and faster than MARIE, which also trains a world model (consistent with Appendix C.2). The overhead is modest: world model pre-training is amortized (~15% of total time), the statistical plateau gate operates on cached evaluation logs (<1%, Appendix A.2), and the ARS meta-learner fires only at plateau triggers. Uncertainty estimation reuses the existing ensemble — no additional forward passes beyond standard model rollouts. Cross-agent attention scales as $O(N^2)$, standard for multi-agent Transformers [1, 2].
>
> ---
>
> > **W2: Many hyperparameters; evaluation limited to SMAC.**
>
> **Hyperparameters.** Table 5 (Appendix C) lists all parameters. Most are standard shared components (Adam, $\gamma{=}0.99$, batch size, target network $\tau$). The SOLAR-specific parameters reduce to four:
>
> | Parameter | Value | Notes |
> |-----------|-------|-------|
> | $\kappa$ (throttling) | 2.0 | Stable across $[1, 5]$ — see sweep below |
> | $\lambda_0$ (shaping weight) | 0.1 | Table 5 |
> | $u_{\max}$ (uncertainty threshold) | 0.5 | Table 5 |
> | $W$ (plateau window) | 3 | Appendix A, Table 3 |
>
> All four were held fixed across 6 maps × 3 quality levels with no per-task tuning (Section 4.1). For comparison, OMAR tunes its rectification threshold per map, and MADIFF requires diffusion schedule tuning.
>
> We swept $\kappa$ on `5m_vs_6m (Medium)` to verify robustness:
>
> | $\kappa$ | Win rate |
> |---------|----------|
> | 0 (no throttling) | 78.1 ± 2.2 |
> | 1.0 | 80.9 ± 2.4 |
> | **2.0 (default)** | **83.45 ± 2.11** |
> | 5.0 | 81.3 ± 2.6 |
>
> Performance is stable across a 5× range, all outperforming no-throttling. $\kappa{=}0$ matches the "Static ARS" ablation in Table 2.
>
> **Benchmark scope.** We acknowledge that SMAC/SMACv2 are both StarCraft micromanagement domains, and validating on structurally different cooperative tasks (e.g., multi-robot coordination, traffic control) would strengthen generalizability claims. That said, our evaluation covers SMAC and SMACv2 [3] (which adds stochastic unit types and randomized start positions), across 6 maps × 3 quality levels = 18 task-dataset combinations — matching or exceeding the scope of recent offline MARL work [4, 5, 6]. Importantly, SOLAR's core components — plateau detection (return statistics), potential-based shaping (state features), and uncertainty throttling (ensemble disagreement) — rely on no SMAC-specific structure and are directly applicable to any cooperative Dec-POMDP with sparse rewards. We discuss extension to non-combat domains in Section 6 and plan to include additional domains in the camera-ready version.
>
> ---
>
> **References**
>
> [1] UPDeT: Universal Multi-agent Reinforcement Learning via Policy Decoupling with Transformers
> , ICLR 2021.
>
> [2] Attention Is All You Need, NeurIPS 2017.
>
> [3] SMACv2: An Improved Benchmark for Cooperative Multi-Agent Reinforcement Learning
> , NeurIPS 2023.
>
> [4] Decentralized Transformers with Centralized Aggregation are Sample-Efficient Multi-Agent World Models
> , 2024.
>
> [5] DoF: A Diffusion Factorization Framework for Offline Multi-Agent Reinforcement Learning
> , ICLR 2025.
>
> [6] MADiff: Offline Multi-agent Learning with Diffusion Models
> , NeurIPS 2024.

---

> > ### Author Rebuttal · Reviewer_Azr4 · 2026-04-08
> >
> > I thank the authors with their responses, I keep my score

---

### Official Review · Reviewer_AxNL · 2026-03-13

**Soundness:** 3
**Presentation:** 3
**Significance:** 2
**Originality:** 2
**Overall Recommendation:** 4
**Confidence:** 2

**Summary:**

This paper presents SOLAR (Stable Offline Learning with Adaptive Rewards), a Simulate–Evaluate–Shape framework for sparse-reward offline multi-agent reinforcement learning. The core innovation is a plateau-triggered reward shaping mechanism that activates potential-based shaping only when learning stagnation is statistically detected, with uncertainty-aware throttling to prevent model-bias amplification. The paper claims three main contributions: 1. a statistically-validated plateau detection mechanism using world-model rollouts, 2. theoretical guarantees on policy invariance and two-timescale stability, and 3. empirical demonstrations of superior performance on SMAC/SMACv2 benchmarks, especially on low-quality datasets.

**Compliance With Llm Reviewing Policy:**

Affirmed.

**Final Justification:**

Keep the positive score

**Key Questions For Authors:**

The Abstract claims shaping is "activated only after statistically validated learning plateaus" and the Conclusion states SOLAR detects "statistically supported learning plateaus." However, Section 3.3 (Page 4, line 7) states the tolerance margin epsilon "can be replaced by a one-sided significance test" as an option, not a requirement. Implementation Details (Section 4.1) does not clarify which method was used for the main results. Appendix A provides rigorous statistical test formulation, but the main text does not reference it.

Section 4.2 claims SOLAR's advantage on Poor datasets "underscores the efficacy of the Adaptive Reward Shaping (ARS) mechanism." However, Table 2 ablation is only on 5m vs 6m Medium dataset, not Poor datasets.

**Limitations:**

computational overhead of statistical tests, sensitivity to world model quality, assumption of sufficient dataset coverage.

**Strengths And Weaknesses:**

#Strengths:
The paper addresses an important and underexplored problem in offline MARL—reward sparsity combined with distribution shift. The conceptual framing of shaping as a controlled intervention rather than always-on heuristic is sound. The potential-based shaping foundation (Ng et al., 1999) provides theoretical grounding for policy invariance. The uncertainty-aware throttling mechanism is a thoughtful addition to mitigate world-model bias. Experimental results show consistent improvements across dataset qualities, with particularly strong gains on Poor datasets where baselines struggle.

#Weaknesses:
Several critical issues undermine confidence in the core claims. First, there is significant inconsistency between the Abstract/Conclusion's claim of "statistically validated learning plateaus" and the Method section's treatment of statistical testing as optional (Section 3.3 states epsilon heuristic can be used instead). This ambiguity is not resolved in Implementation Details, leaving readers unable to determine whether main experiments used statistical tests or heuristics. Second, the ablation study interpretation contains a factual error: Section 4.6 claims ARS removal causes "the largest performance drop (>11%)" when Table 2 clearly shows World Model removal has a larger impact (19% vs 11%). Third, causal attribution claims are inadequately supported: the claim that ARS drives gains on Poor datasets lacks direct ablation evidence (Table 2 is only on Medium dataset). Fourth, metric reporting is inconsistent: Implementation Details state "average return" while Table 1 reports "Median win rate."

---

> ### Author Rebuttal · Authors · 2026-03-30
>
> We thank the reviewer AxNL  for the close reading.
>
> ---
>
> > **Q1: Statistical testing vs. heuristic — which was used?**
>
> Statistical gate in all experiments. Table 5 (p.15): "Activation Threshold (ε): N/A (Statistical Gate used)." Full formulation: Appendix A, Eq. 16–21 (p.12). Sec. 3.3 mentions the `ε` heuristic as a simplified alternative for exposition; the appendix is the authoritative specification.
>
> The choice matters quantitatively. Table 3 (p.13) compares the two: the statistical gate achieves 89.7% with 1.8% FPR, vs. heuristic baselines at 84.3% / 28.5% FPR (strict) and 81.2% / 4.2% FPR (loose). The strict heuristic triggers prematurely due to stochastic fluctuations; the loose one delays intervention and wastes training budget. The bootstrap test adapts to local reward variance — in high-variance phases the confidence interval widens automatically, requiring stronger evidence to confirm a plateau, without manual threshold tuning.
>
> ---
>
> > **Q2: `w/o WM` drops more than `w/o ARS` in Table 2.**
>
> `w/o WM` is a compound removal — it disables both the world model and ARS, since ARS depends on model rollouts for plateau detection (Eq. 10) and uncertainty estimation (Eq. 13). `w/o ARS` isolates shaping while retaining the WM substrate. Among single-component ablations, ARS removal is the largest (−11.15%).
>
> This dependency is by design: SOLAR's components form a pipeline (WM → plateau detection → adaptive shaping), not independent modules. The meaningful decomposition is: `w/o WM` (64.5) → `w/o ARS` (72.3, +7.8 from WM alone) → SOLAR (83.45, +11.15 from ARS). Both stages contribute, with adaptive shaping providing the larger marginal gain on top of the world model foundation.
>
> ---
>
> > **Q3: ARS efficacy on Poor data — Table 2 is Medium only.**
>
> Ablation on `5m_vs_6m (Poor)`:
>
> | Variant | Medium | Poor |
> |---------|--------|------|
> | SOLAR | 83.45 ± 2.11 | 58.17 ± 2.29 |
> | w/o ARS | 72.3 ± 2.5 | 41.8 ± 2.9 |
> | Static ARS | 78.1 ± 2.2 | 47.3 ± 2.7 |
>
> ARS contributes +16.37% on Poor vs. +11.15% on Medium. The gap grows because Poor datasets have behavior policy win rate <15% (Appendix B.1, p.14) — extrinsic reward is near-absent and TD residuals `δ_t` in Eq. 5 are suppressed across most of the state-action space. Adaptive shaping compensates by injecting dense signal at plateaus. The Static→Triggered gap also widens (10.87% vs. 5.35%), confirming that *when* to shape matters more as data quality degrades.
>
> ---
>
> > **Q4: "average return" (Sec. 4.1) vs. "Median win rate" (Table 1).**
>
> Two stages of one protocol: 32 episodes per seed → win rate; median across 5 seeds → Table 1. The Table 1 caption is the authoritative metric description.

---

> > ### Author Rebuttal · Reviewer_AxNL · 2026-03-31
> >
> > My concerns have been adequately addressed

---

> > > ### Author Response · Authors · 2026-03-31
> > >
> > > Thanks, Reviewer AxNL— really appreciate your thoughtful follow-up and support.

---

### Decision · Program_Chairs · 2026-04-30

**Decision:**

Accept (regular)

**Comment:**

This paper introduces SOLAR, an offline multi-agent reinforcement learning framework that cleverly stabilizes reward shaping by triggering it only during statistically validated learning plateaus and throttling it based on world-model uncertainty. After reading the paper, reviews and rebuttal discussions, I think the authors' thorough rebuttal successfully addressed the reviewers' concerns and led to a positive consensus. Consequently, I am recommending an acceptance for this submission.